# Validation of precipitation reanalysis products for rainfall-runoff modelling in Slovenia

Marcos Julien Alexopoulos[1,2], Hannes Müller-Thomy[3,*], Patrick Nistahl[3], Mojca Šraj[4], Nejc Bezak[4]

[1] EMVIS S.A., Consultant Engineers-Environmental Services, Research Information Technology & Services, 15343 Athens, Greece

[2] National Technical University of Athens, Department of Water Resources and Environmental Engineering, School of Civil Engineering, 15780 Athens, Greece

[3] Technische Universität Braunschweig, Leichtweiß-Institute for Hydraulic Engineering and Water Resources, Division of Hydrology and River Basin Management, Germany

[4] University of Ljubljana, Faculty of Civil and Geodetic Engineering, Jamova cesta 2, Ljubljana, Slovenia

[*] previously published under the name Hannes Müller

*Correspondence to: Hannes Müller-Thomy (h.mueller-thomy@tu-braunschweig.de)*

*OrcIDs:*

| | |
|---|---|
| Marcos Julien Alexopoulos | 0000-0002-7164-8547 |
| Hannes Müller-Thomy | 0000-0001-5214-8945 |
| Patrick Nistahl | 0000-0001-9000-8249 |
| Mojca Šraj | 0000-0001-7796-5618 |
| Nejc Bezak | 0000-0003-2264-1901 |

**Abstract.** Observational data scarcity often limits the potential of rainfall-runoff modelling around the globe. In ungauged catchments, earth-observations or reanalysis products could be used to replace missing ground-based station data. However, performance of different datasets needs to be thoroughly tested, especially at finer temporal resolutions such as hourly time steps. This study evaluates the performance of ERA5-Land and COSMO-REA6 precipitation reanalysis products (PRPs) using 16 meso-scale catchments (41-460 km²) located in Slovenia, Europe. These two PRPs are firstly compared with a gridded precipitation dataset that was constructed based on ground observational data. Secondly, a comparison of the temperature data of these reanalysis products with station-based air temperature data is conducted. Thirdly, several data combinations are defined and used as input for the rainfall-runoff modelling using the GR4H model. A special focus is on the application of an additional snow module. Both tested PRPs underestimate, for at least 20%, extreme rainfall events that are the driving force of natural hazards such as floods. In terms of air temperature both tested reanalysis products show similar deviations from the observational dataset. Additionally, air temperature deviations are smaller in winter compared to summer. In terms of rainfall-

runoff modelling, the ERA5-Land yields slightly better performance than COSMO-REA6. If a re-calibration with PRP has

been carried out, the performance is similar compared to the simulations where station-based data was used as input. Model recalibration proves to be essential in providing relatively sufficient rainfall-runoff modelling results. Hence, tested PRPs could be used as an alternative to the station-based data in case that precipitation or air temperature data are lacking, but model calibration using discharge data would be needed to improve the performance.

**Keywords**

Reanalysis products; precipitation; rainfall-runoff modelling; temperature; Slovenia

**1 Introduction**

High quality high-resolution observations of atmospheric variables are of crucial importance for hydrological applications.

Insightful approximations of catchment behavior are heavily dependent on the availability and accuracy of precipitation and temperature records. Given that most catchments around the world display a significant lack of weather station coverage, a plethora of methods have been developed to deal with data scarcity (e.g., remote sensing measurements, data-assimilated gridded products, general circulation models). Amongst them, an option that has been increasingly popular in recent years, is the use of reanalysis products (Onogi et al., 2007; Rienecker et al., 2011; Gelaro et al., 2017), which is based on meteorological

models that assemble surface observations, but mostly data gathered by means of remote sensing technology. That is, remote sensing observations are assimilated in the dynamic model to guide the simulation of the reanalysis data. This provides the advantage of producing information at multiple vertical atmospheric levels (Muñoz-Sabater et al., 2021; Vousdoukas et al., 2016; Ruane et al., 2015; Marques et al., 2009), in addition to providing coverage regardless of the status of the surface observational network. Due to the combination of multiple observational data sets as input the resulting reanalysis data can

outperform individual observational data sets, as shown by e.g. Gebremichael et al. (2017) and Zhang et al. (2021) for satellite data. Also, reanalysis data typically provide better temporal coverage compared to datasets derived via remote sensing techniques. For example, ERA5-Land (Muñoz-Sabater et al., 2021; Hersbach et al., 2020) is currently providing information from January 1950 onwards, and at the time of writing, is expected to be extended to cover a timespan starting from 1940, providing an even longer period for land surface variable data. Thus, if accurate, they provide a valuable resource for studying

climate variability, long-term trends, and their impacts on hydrological processes, such as droughts or floods. On the spatial scale, reanalysis datasets offer spatially consistent precipitation fields that account for regional variability and orographic effects. Plus, there are studies showing that reanalysis is superior to satellite precipitation, as in e.g., Ougahi and Mahmood, 2022. However, before applications, proper validation is necessary, as each product is tailored for different regions worldwide. Several studies on inter-comparisons between various reanalysis products exist in order to identify their suitability for a

particular region (Koohi et al., 2022). Lauri et al. (2014) made an evaluation of bias-corrected ERA-Interim (Dee et al., 2011) and Climate Forecast System Reanalysis (CFSR) (Saha et al., 2010, 2014) precipitation and temperature in the Mekong catchment in southeast Asia, for the period 1999-2005. The spatial pattern of Era-Interim temperature displays greater

resemblance to observations compared to CFSR. However, the difference between daily maximum and minimum temperature proves to be more realistic for CFSR. Average annual rainfall is similar for all datasets, however CFSR tends to overestimate rainfall at the lower-middle part of the study area. Islam and Cartwright (2020) evaluated the performance of the European Centre for Medium-Range Weather Forecasts (ECMWF) Reanalysis V5 (ERA5) (Hersbach et al., 2020) and CFSR precipitation products in Bangladesh over a 5-year period, with the resolution aggregated at the daily scale. CFSR tends to overestimate rainfall patterns across 90% of the domain. ERA5 tends to overestimate rainfall for over 50% of the area, while still performing reasonably well. However, above the $50^{th}$ and the $75^{th}$ percentiles of rainfall records, it shows an underestimation of 49% and 85%, respectively, in contrast to CFSR. The study also evaluated the ability of the products to detect rainfall. Using the Probability of Detection (POD) and Volumetric Hit Index (VHI) metrics, both datasets display superior performance in detecting the occurrence of rainfall, with CFSR outperforming ERA5 for higher rainfall values. The number of false alarms was also evaluated using the False Alarm Ratio (FAR), where CFSR displays the poorest performance, especially for higher rainfall thresholds. Jiang et al. (2021) evaluated the performance of ERA5 precipitation for a 12-year period over Chinese mainland. The results detect an optimal rainfall detection capacity, but a tendency to overestimate total precipitation while underestimating heavy rainfall events, which is consistent with other recent findings (Hénin et al., 2018; Beck et al., 2019; Sharifi et al., 2019; Amjad et al., 2020; Xu et al., 2019; Nogueira, 2020; Mahto and Mishra, 2019). At a smaller scale, Khan et al. (2020) assessed the application of the Japanese Reanalysis (JRA-55) (Kobayashi et al., 2015) and ERA-Interim precipitation for the Pindiali, Dande and Sarobi dams, in the Khyber-Pakhtunkhwa province of Pakistan. On a monthly average basis, both products show great rainfall overestimation for the period 1979-2010, during both wet and dry seasons.

The potential of reanalysis precipitation has also been investigated in rainfall-runoff simulations. Wang et al. (2020) tested the efficiency of the China Meteorological Assimilation Driving Datasets (CMADS) (Meng et al., 2019) and CFSR in the Xihe river catchment in China. In terms of precipitation performance at the catchment scale, CMADS tends to underestimate mean precipitation compared to observations, especially during wet season. CFSR shows a great overestimation, with approximations of annual rainfall differing by about 80%. In rainfall detection, CMADS displays adequate skill in capturing rainfall events, in addition to acceptable FAR results. According to the POD metric, CFSR performs rather poorly in detecting rainfall, contradicting Islam and Cartwright (2020). The aforementioned products were used as an input in the Soil & Water Assessment Tool (SWAT). Simulations were performed at the monthly scale from 01/2009 till 12/2015. The use of the CFSR dataset proved to be inadequate and was discarded as an option, while CMADS resulted in a large runoff underestimation. Hafizi and Sorman (2021) evaluated the performance of ERA5 precipitation in the Karasu catchment in eastern Turkey, over the period 2014-2019 at a daily time step. Overall, the product shows high detectability for low and moderate precipitation, regardless of seasonality. In terms of streamflow reproducibility, the simulations perform weakly when the model parameters are calibrated using observed data. When calibrated individually, flow reproducibility is high for both calibration and validation periods. Ghodichore et al. (2018) addressed the applicability of the APHRODITE (Yatagai et al., 2009, 2012), ERA-Interim, PERSIANN (Hsu et al., 1997; Sorooshian et al., 2000) and TMPA-RT (Huffman et al., 2007) reanalysis products over the

Sefidrood catchment in Iran, at the daily and monthly timestep. At the latter, all products perform in a similar fashion, with APHRODITE performance slightly exceeding the rest of the selection. When increasing temporal resolution, APHRODITE and ERA-Interim are better able to capture rainfall station measurements. Feng et al. (2021) evaluated precipitation reanalysis

in the United States. Their comparative analysis shows mixed results: reanalysis is unable to capture rainfall dynamics in the northern part, whilst results are adequate in the rest of the study area. Another study using SWAT assessed the performance of 14 remote sensing products over a macro-scale watershed in Pakistan. Amongst 14 satellite-, gauge- and reanalysis precipitation, APHRODITE and JRA-55 are the most adequate in capturing rainfall dynamics at the daily scale (Saddique et al., 2022). Not much has been investigated for snowmelt-driven runoff, however Bhattacharya et al. (2019) suggest that

reanalysis datasets can outperform observations at the monthly scale.

Overall, for reanalysis datasets, research is mostly focused on the evaluation of precipitation, usually derived at a coarse spatial and temporal resolution. Fewer publications have focused on the validity of air temperature in addition to precipitation, especially within rainfall-runoff modelling applications. In the research studies where this focus was put, most reanalysis datasets are subject to some bias-correction adjustment before further use. At the time of writing and according to the best of

the authors knowledge, no rainfall-runoff validation has been made for non-bias corrected precipitation and air temperature products on European catchments at the hourly time step. In addition, a multi-catchment analysis has yet to be conducted, where correlations can be made between reanalysis performance on streamflow simulation and different catchment characteristics.

Therefore, the main objective of this study is to evaluate the potential of ERA5-Land ($\Delta t$=1h, $\Delta l$=0.1°, Muñoz-Sabater et al.,

2021) and COSMO-REA6 ($\Delta t$=1h, $\Delta l$=0.055°, Bollmeyer et al., 2015) as raw, non-bias adjusted precipitation and air temperature reanalysis products for 16 catchments in Slovenia, Central Europe. These two products were selected, since their spatial and temporal resolution seems to be sufficient to cope with the dynamics of rainfall-runoff processes modelled. Initially, a comparison of precipitation and air temperature is conducted against weather station observations using various performance metrics. The ability of these products to detect precipitation is also investigated. Furthermore, using a conceptual rainfall-

runoff model, an evaluation of discharge simulations is performed against measurements and several important conclusions are drawn. Therefore, selected catchments are considered as ungauged in terms of precipitation and air temperature despite the fact that ground-based data is available in Slovenia. It will be analysed if the good spatial and temporal coverage of the reanalysis data can compensate possible quantitative deviations of rainfall amounts from the observational station network, which comes therefore with less spatial and temporal coverage.

**2 Data**

**2.1 Catchment characteristics**

Sixteen Slovenian catchments are selected for the present case study, representing the five different discharge regimes in Slovenia (Frantar et al., 2008; Frantar and Hrvatin, 2008). A spatial representation of the catchments is displayed in Fig. 1. Table 1 illustrates the main catchment characteristics. Catchments with larger areas typically receive more precipitation and

have higher water storage capacities than smaller catchments, due to larger surface covered. This leads to a lower runoff coefficient and slower runoff response, as water moves through the catchment over longer distances and time periods. Median elevation is another important factor that influences the generation of runoff. Catchments with higher median elevations generally experience higher precipitation amounts as a result of orographic uplift, which forces moist air to rise and cool, leading to enhanced condensation and precipitation. The percentage of forest area is also a significant factor that affects the

generation of runoff. Forested catchments generally have lower runoff coefficients and slower runoff response due to the high rainfall interception rates, and high-water storage capacities of forest soils. The presence of trees also reduces the erosive power of runoff, which leads to lower sediment yields and improved water quality. Mean catchment slope is another factor that contributes in the generation of runoff. Catchments with steeper slopes have higher runoff coefficients and more rapid runoff response due to the reduced infiltration capacity of the soils, and the rapid movement of water down the slope. In

contrast, catchments with flatter slopes have lower runoff coefficients and slower runoff response due to the higher infiltration capacity of the soils and the slower movement of water across the landscape. Vis-à-vis the discharge regimes, Alpine nivo-pluvial regimes occur in catchments whose greater part reach into high mountains, where snow melt effects are especially pronounced in May/June, while Alpine pluvio-nival regimes describe water behavior for catchments located in the medium height of Alpine mountains. Lahinja, Bolska-Dolenja vas (Dvas) and Idrijca-Hotešk rivers comprise the Dinaric area and

follow a Dinaric pluvio-nival regime, where discharge peaks occur during spring and autumn. The rivers flowing through the hills of the Pannonian area are described by early summer and late autumn peaks which are strongly equalised, exhibiting low flows mainly during the summer. Catchments located in the south-western part of Slovenia show a Mediterranean pluvial regime with main peaks occurring during the months of November and December, with decreased water levels observed in August.

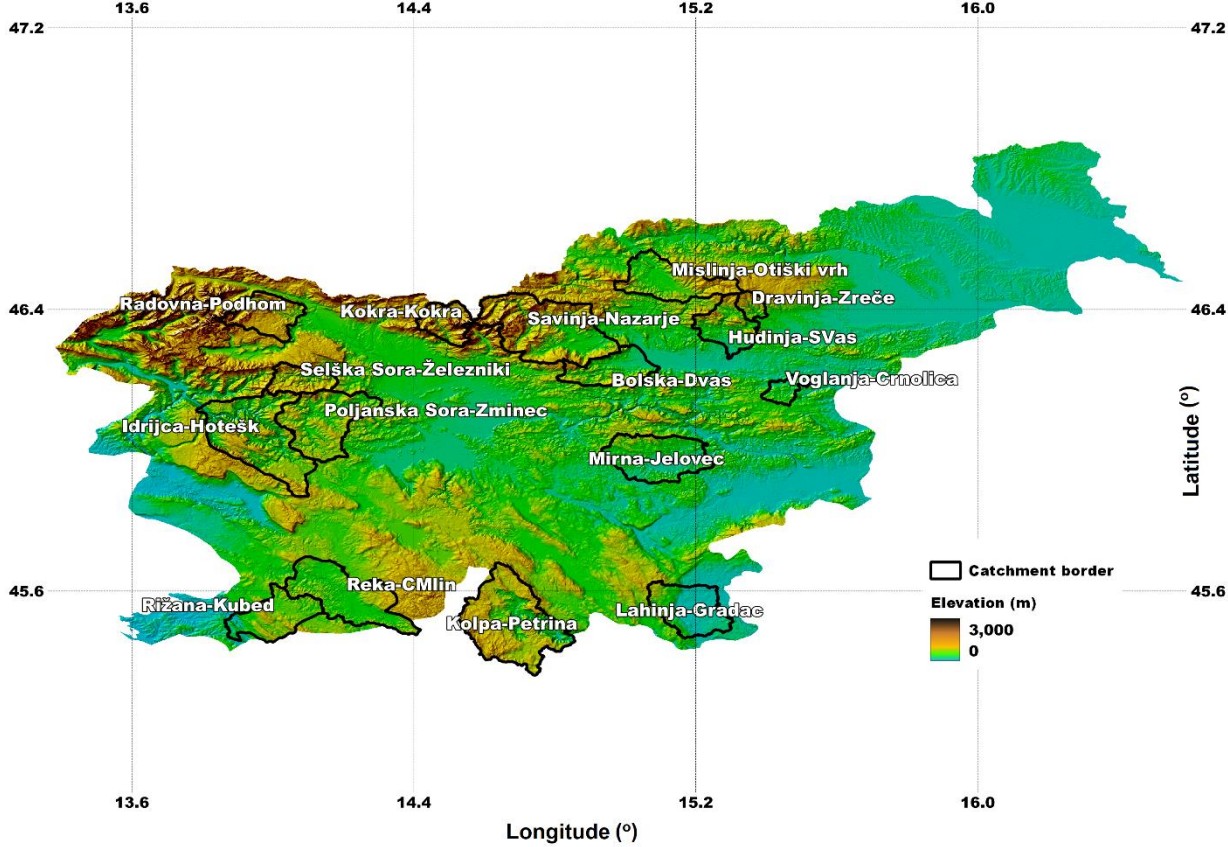


**Figure 1: Catchment location on the map of Slovenia with the elevation background.**

**Table 1: Main catchment characteristics.**

| River-Gauging station | Centroid coordinates (lon/lat) – WGS84 | Average annual rainfall (mm) | Catchment area (km$^2$) | Median elevation (m.a.s.l.) | Percentage of forest cover (%) | Mean catchment slope (°) | Discharge regime |
|---|---|---|---|---|---|---|---|
| Mislinja-Otiški vrh | 15.04/46.56 | 1130 | 230 | 950 | 66 | 15 | Alpine pluvio-nival |
| Dravinja-Zreče | 15.38/46.38 | 1007 | 41 | 972 | 68 | 16 | Pannonian pluvio-nival |
| Radovna-Podhom | 14.08/46.39 | 2336 | 166 | 1556 | 94 | 19 | Alpine nivo-pluvial |

| | | | | | | | |
|---|---|---|---|---|---|---|---|
| Kokra-Kokra | 14.49/46.30 | 1682 | 112 | 1561 | 94 | 27 | Alpine pluvio-nival |
| Poljanska Sora-Zminec | 14.29/46.15 | 1492 | 305 | 945 | 66 | 15 | Alpine pluvio-nival |
| Selška Sora-Železniki | 14.16/46.22 | 1746 | 104 | 1065 | 84 | 22 | Alpine pluvio-nival |
| Mirna-Jelovec | 15.23/45.98 | 1242 | 270 | 530 | 57 | 10 | Pannonian pluvio-nival |
| Kolpa-Petrina | 14.85/45.46 | 1590 | 460 | 863 | 88 | 14 | Alpine pluvio-nival |
| Lahinja-Gradac | 15.24/45.61 | 1371 | 221 | 593 | 73 | 5 | Dinaric pluvio-nival |
| Savinja-Nazarje | 14.95/46.32 | 1012 | 457 | 1344 | 81 | 21 | Alpine pluvio-nival |
| Bolska-Dolenja vas (Dvas) | 15.09/46.23 | 1022 | 175 | 876 | 63 | 14 | Dinaric pluvio-nival |
| Voglanja-Črnolica | 15.41/46.19 | 1170 | 53 | 470 | 37 | 11 | Pannonian pluvio-nival |
| Hudinja-Škofja vas (SVas) | 15.28/46.26 | 1106 | 156 | 875 | 57 | 14 | Alpine pluvio-nival |
| Idrijca-Hotešk | 13.79/46.12 | 1545 | 442 | 831 | 79 | 19 | Dinaric pluvio-nival |
| Reka-Cerkvenikov mlin (Cmlin) | 14.06/45.65 | 1331 | 377 | 801 | 70 | 9 | Mediterranean pluvial |
| Rižana-Kubed | 13.87/45.53 | 1003 | 204 | 554 | 80 | 9 | Mediterranean pluvial |

Figure 2 displays the annual mean, minimum and maximum flows averaged over the years 2009-2014, for which rainfall-runoff simulations are performed (further details are provided in sub-section 3.3.2). Overall, alpine pluvio-nival catchments

that cover areas greater than 200 km$^2$, such as Poljanska Sora-Zminec, Kolpa-Petrina, Savinja-Nazarje and Idrijca-Hotešk, tend to demonstrate higher maximum discharge regimes, ranging between 1-5 mm/h. Additionally, rivers located towards the eastern part of the country exhibit lower flow rates (Voglanja-Črnolica or Bolska-Dolenja vas (Dvas)), deviating almost by

two orders of magnitude from their yearly mean.

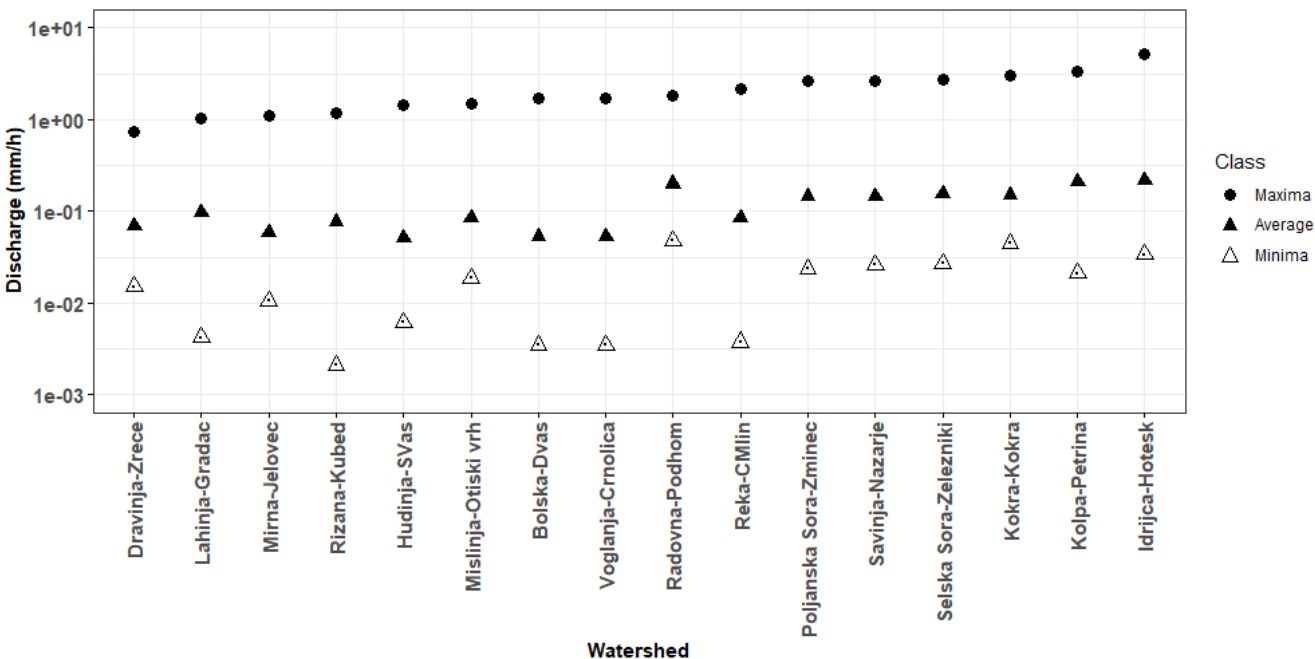

**Figure 2: Average values of annual mean, minimum and maximum discharge values per catchment (mm/h). Rivers (catchments) are displayed in ascending order of maximum discharge from left to right. Y axis is logarithmically scaled.**

**2.2 Observed data**

For the validation of the precipitation reanalysis products (PRP), a regionalized daily precipitation data set from the Slovenian Environment Agency (ARSO) is used, from now on referred to as ARSO-d. ARSO-d has a spatial resolution of 1 km raster width and length and is available from 01.01.1981 to 31.12.2010. It is based on the regionalization and upscaling of station-based precipitation measurement into spatially and temporally consistent dataset. For the rainfall-runoff modelling, hourly

precipitation, air temperature and discharge measurements are obtained for the period 2009-2014, also provided by ARSO. To set up observational time series for precipitation and air temperature, one representative station is selected per catchment. The selection of each representative precipitation and air temperature station is derived based on its proximity to the respective catchments' centroid. Once selected, a correlation analysis between the representative stations and stations within a radius of 15 km is conducted. Stations with a correlation coefficient below 0.6 are screened out. Remaining stations are classified in a

descending order based on the previously calculated correlation coefficient. To account for missing values of each

representative station, values are borrowed by the neighbouring station with the highest correlation, and if missing, by the station with the second-best correlation, etc. Furthermore, the borrowed values are transformed by following a linear regression scheme between the two stations. Selected representative stations of individual catchments are shown in Figure 3. Due to limited availability, some catchments are set up by using the same stations (e.g., Poljanska Sora-Zminec, Selška Sora-Železniki, Idrijca-Hotešk). In addition, some representative stations (e.g., Voglanja-Črnolica, Rižana-Kubed) are located outside of the catchment boundary, which could hinder their calibration process by failing to represent the dynamics of the area due to the spatial variability of precipitation.

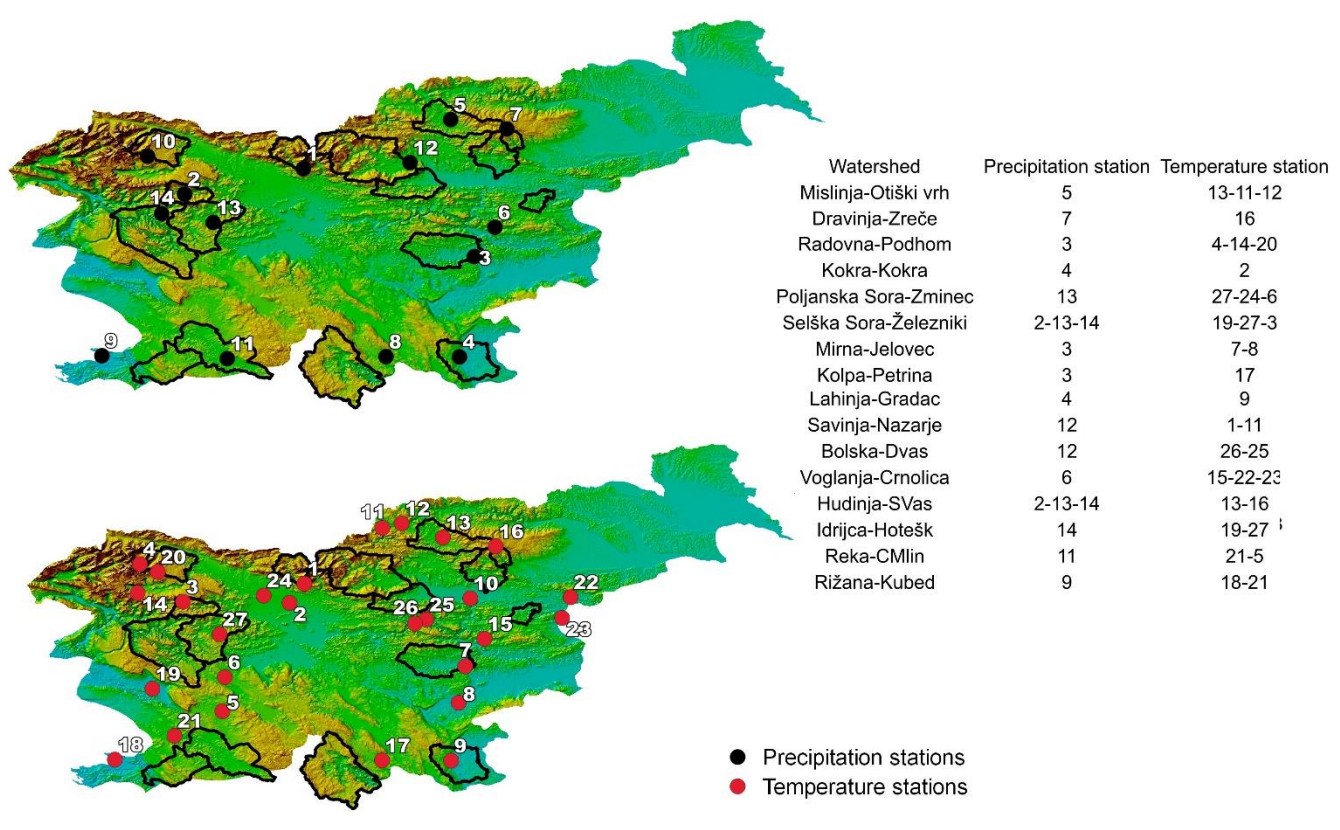

| Watershed | Precipitation station | Temperature station |
|---|---|---|
| Mislinja-Otiški vrh | 5 | 13-11-12 |
| Dravinja-Zreče | 7 | 16 |
| Radovna-Podhom | 3 | 4-14-20 |
| Kokra-Kokra | 4 | 2 |
| Poljanska Sora-Zminec | 13 | 27-24-6 |
| Selška Sora-Železniki | 2-13-14 | 19-27-3 |
| Mirna-Jelovec | 3 | 7-8 |
| Kolpa-Petrina | 3 | 17 |
| Lahinja-Gradac | 4 | 9 |
| Savinja-Nazarje | 12 | 1-11 |
| Bolska-Dvas | 12 | 26-25 |
| Voglanja-Crnolica | 6 | 15-22-23 |
| Hudinja-SVas | 2-13-14 | 13-16 |
| Idrijca-Hotešk | 14 | 19-27 |
| Reka-CMlin | 11 | 21-5 |
| Rižana-Kubed | 9 | 18-21 |

**Figure 3: Location of representative precipitation and temperature stations together with selected catchments used within this study.**

### 2.3 Reanalysis products

This study evaluates the performance of two precipitation and temperature reanalysis products. ERA5-Land (Muñoz-Sabater et al., 2021) is the 5[th] generation climate reanalysis dataset produced by ECMWF. Considered the ERA-Interim successor, it

holds substantial upgrades with a finer spatial scale and temporal resolution. The atmospheric variables are driven by the
simulation that is subsequently corrected by a four-dimensional variational assimilation scheme (4D-Var) (Courtier et al.,
1994; Bonavita et al., 2016) that exploits observations gathered by conventional and remote sensing instruments. COSMO-
REA6 is a regional reanalysis product that covers the CORDEX domain (Bollmeyer et al., 2015). The simulation follows a
continuous nudging scheme (Bollmeyer et al., 2015; Stephan et al., 2008) to allow the continuous assimilation of observations.
Summary information of the reanalysis data is shown in Table 2. Points from the reanalysis grid cells were acquired based on
their spatial overlap with the respective catchment's centroid.

**Table 2: Main attributes of the precipitation and air temperature products.**

| Product | Spatial coverage | Period | Spatial resolution | Temporal resolution | Vertical levels | Reference |
|---|---|---|---|---|---|---|
| ERA5-Land | Global | 1950-Present | $0.1° \times 0.1°$ | 1 h | 137 | Muñoz-Sabater et al., 2021 |
| COSMO-REA6 | Europe | 1995-2019 | $0.055° \times 0.055°$ | 1 h | 40 | Bollmeyer et al., 2015 |

## 3 Methods

### 3.1 Validation of precipitation reanalysis products

A comparison of the PRP with rain gauge data is not carried out, because it would be potentially invalid due to the different
spatial resolution of both datasets. Rain gauge data is the most representative for the catching area of the measuring instrument
(e.g., 200 cm² for the well-known Hellmann measuring instrument), while the PRP data represent approx. 36 km² (COSMO-
REA6) and approx. 81 km² (ERA5-Land). PRP will be longer and more frequently affected by a storm due to the represented
area (higher wet spell duration, more wet spell events, smaller probability of dry time steps), which also leads to a smoothing
of the rainfall process in space (smaller rainfall intensities, especially for extreme values of short durations).
Hence, for the validation of the PRP the ARSO-d data is used, because it is the only available data set with a sufficient spatial
coverage to enable more robust comparisons with the PRP. Unfortunately, it is only available at the daily time step. The
validation is carried out at the catchment scale (Table 1) and only the overlapping data periods are used. This overlapping
period is chosen for each PRP separately to ensure the highest possible time series length for comparisons. The resulting
validation periods are for ERA5-Land: 1.1.1981-31.12.2010 and for COSMO-REA6: 1.1.1995-31.12.2010. The PRP are
aggregated to daily values to enable the comparison with ARSO-d, since no hourly spatial rainfall product for Slovenia is

provided by ARSO. The validation is carried out using the relative error rE between observations (Obs) and reanalysis data for each studied precipitation characteristic (RC) as mean value over all n stations:

$$rE = \frac{1}{n} \times \sum_{i=1}^{n} \frac{(RC_{PRP,i} - RC_{Obs,i})}{RC_{Obs,i}} \qquad (1)$$


As RC event and continuous characteristics as well as extreme values are analysed (as e.g., in Pohle et al., 2018, Müller-Thomy, 2019, 2020). Events are defined as wet time steps enclosed by at least one dry time step before and afterwards, to derive RC as dry spell duration, wet spell duration and wet spell amount. Continuous RC are average intensity and probability of dry intervals. For the extreme values, peak-over-threshold series of precipitation extreme values are extracted with three events per year on average (DWA-A 531, 2012).


### 3.2 Validation of air temperature reanalysis products

For the validation of air temperature time series of reanalysis products, the maximum available period for all hourly station-based temperature data sets is applied: 01.01.2009-31.08.2019. For each catchment, the corresponding air temperature station is used as shown in Figure 3. For the entire period, median temperature values for each day are derived for catchment-specific analyses. For comparison of air temperature series among the catchments monthly median values are derived. For the quantification the absolute error (aE) is used, which is temperature difference between the reanalysis product in comparison to the station-based air temperature values (Figure 3), set for a certain period:


$$aE = RC_{PRP} - RC_{Obs} \qquad (2)$$


### 3.3 Validation of reanalysis products using rainfall-runoff modelling

In the current study, the hydrological utility of reanalysis products is additionally evaluated with the use of the lumped conceptual Génie Rural à 4 paramétres Horaires (GR4H), and Génie Rural à 4 paramétres Horaires Cema Neige models. The GR4H model is based on the three-parameter version of the Genie Rural Journalier (GRJ) model, developed by Perrin (2002), scaled to an hourly time step, with the aim of simulating rainfall-runoff by introducing the least amount of parameters. The variables used in the conceptual model are precipitation (*P*) and potential evapotranspiration *(E)*. *E* is a function of surface temperature (*T*) and can be calculated at an hourly time step using the Oudin formula (Oudin et al., 2005). More details about the formula can be found in https://webgr.inrae.fr/en/models/evapotranspiration-model/, where a workbook template is provided for the estimation of *E* in Excel spreadsheet format. Four parameters are ingrained in the GR4H model: *X1* represents the maximum capacity of the production store (i.e. upper reservoir shown in Figure 4) (mm); *X2* is the groundwater exchange coefficient (mm) (i.e. exchange with the lower reservoir shown in Figure 4); *X3* accounts for the one day ahead maximum capacity of the routing store (lower reservoir in Figure 4) (mm); and *X4* is the unit hydrograph time base (used to derive the Unit Hydrograph 1 (UH1) and Unit Hydrograph 2 (UH2) as shown in Figure 4).



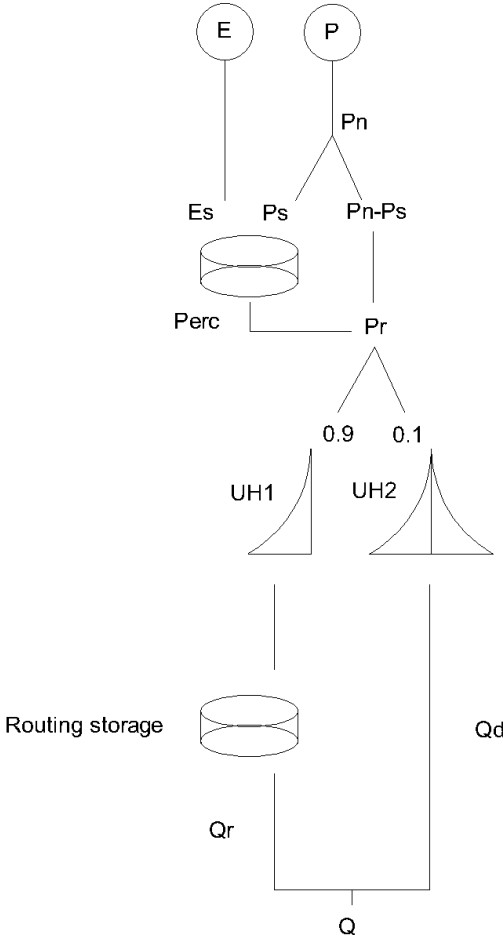

**Figure 4: A schematic representation of the GR4H model (adopted after Perrin et al., 2003).**

*P* and *E* data is used to calculate net rainfall (*Pn*), which is then used to fill the production store (*Ps*) and to perform run-off routing (*Pn−Ps*) (Figure 4). The production store is emptied by percolation (*Perc=f(S, X1)*, where *S* is the production store level) or by the rate of potential evapotranspiration (*Es=f(S, X1, En)*, where *En* the net evapotranspiration capacity) (Figure 4). The difference between net rainfall and rainfall that is used to fill the production store (*Pn−Ps*) is then used together with percolation from the production store (*Perc*) to calculate flow (*Pr*). Multiple routing steps are then applied to simulate flow values (Figure 4). *Pr* is divided into two parts, 90% is being routed by the unit hydrograph UH1 (*X4*) and a routing store (*X3*) while 10% is routed by the unit hydrograph UH2 (*X4*) (Figure 4). In the case of the UH2 and the routing store, a groundwater exchange term (gain or loss) is also introduced (*X2* parameter). Further details about the lumped conceptual rainfall-runoff model can be found in Perrin et al. (2003).

The Cema Neige model is a semi-distributed Snow Accounting Routine (SAR) implementing a snowmelt factor and a cold-content factor. The inputs required are $P$ and $T$. For modelling purposes at the catchment scale, the catchment is divided into five elevation zones of equal area. On each elevation band and for each time step, the five functions described in Valery (2010); Valéry, Andréassian and Perrin (2014a, 2014b) are executed in order to compute rain and snowmelt. The outputs from each elevation zone are averaged with an equal weight and used as an input in the GR4H module. Solid precipitation is calculated by multiplying average yearly rainfall on each catchment, with the catchment's percentage of snowmelt. The percentage of snowmelt (in relation to total annual precipitation) is calculated by using an empirical equation derived from data gathered at the daily scale over the precipitation network (ARSO), for the period 2010-2016 (percentage of snowmelt = 0.0168*ME + 3.5128 where ME is the mean catchment or station elevation).

The simulation period is split-sampled into: i) a calibration period (2009-01-01 01:00:00 – 2012-01-01 12:00:00) using a warm-up period of one year (2008-01-01 00:00:00 – 2009-01-01 00:00:00), and ii) a validation period (2012-01-01 13:00:00 – 2014-12-31 23:00:00), with a warm-up period of four years (2008-01-01 00:00:00 – 2012-01-01 12:00:00). It is very well established that split-sampling is recommended if both calibration and validation periods represent similar climate, soil properties and land cover conditions, i.e. consistent catchment conditions over time. Nonetheless, in the data used for this study there are minimal fluctuations within the selected periods in terms of very wet or dry periods. In addition, amongst the various methodologies for calibration/validation period selection found in literature, some studies support the split-sampling approach (e.g., Perrin et al., 2003). Therefore, a 'classical' split-sampling approach was implemented. However, other methodologies could be tested in future studies. Model runs are performed using the following data configurations as input for both simulation periods as shown in Table 3. The purpose of the table is to indicate the data type (temperature, precipitation) and data set (observations, ERA5-Land, COSMO-REA6) used for each combination (1 – 9), rather than to present specific values. Therefore, the symbols (box, circle) are used to represent the data sources instead of actual values.

**Table 3: Data set combinations of precipitation (○) and temperature (□) used for r-r modelling. Colours used are the same as in the following figures.**

| Data origin | 1 | 2 | 3 | 4 | 5 | 6 | 7 | 8 | 9 |
|---|---|---|---|---|---|---|---|---|---|
| Observations | ○□ | □ | □ | ○ | ○ | | | | |
| ERA5-Land | | ○ | | □ | | ○□ | | ○ | □ |
| COSMO-REA6 | | | ○ | | □ | | ○□ | □ | ○ |

Four model runs are performed for each simulation period. The GR4H and GR4H Cema Neige modules are used twice. Initially, the parameters ingrained within the model (*X1, X2, X3, X4)* are calibrated using configuration 1, and used for the remaining eight data configurations (i.e., combinations 2-9 as shown in Table 3). Then, simulations are repeated for a second

time, implementing the Michel calibration algorithm (Michel, 1991) for each data configuration (i.e. combinations 2-9), in order to further evaluate the applicability of the ERA5-Land and COSMO-REA6 datasets within the rainfall-runoff model used in the current study. The purpose of this experiment is to identify whether parameters ingrained within the rainfall-runoff (r-r) model can make up for deficiencies in the reanalysis forcing. In case they do, then subpar performance can be directly attributed to catchment characteristics (e.g., location, elevation, slope patterns).

As a means of performance, the KGE metric is used (Gupta et al., 2009; Kling et al., 2012), which is a combination of bias, variability ratio and correlation. Just like other performance metrics (Nash and Sutcliffe, 1970), KGE = 1 suggests perfect agreement between observations and simulations. According to some authors (Koskinen et al., 2017; Castaneda-Gonzalez et al., 2018), KGE < 0 indicates that the mean of observation provides better estimates than the simulated mean, while others consider negative KGE values simply undesirable (Andersson et al., 2017; Fowler et al., 2018; Siqueira et al., 2018). Knoben

et al. (2019) pointed out that mean flow as benchmark is already outperformed with KGE values >-0.41.

## 4 Results and discussion

### 4.1 Validation of precipitation reanalysis products

The results for the PRP validation using precipitation characteristics are shown in Table 4, Figure 5 and Figure 6. During the

validation of the PRP, a high fraction of very small rainfall intensities is identified for ERA5-Land. These very small rainfall intensities have no relevant impact on the rainfall-runoff process, especially not for extreme floods and flood frequency analysis. The same can be said for the soil erosion due to water. Hence, to get a more representative validation of the PRP, thresholds of 0.01 mm, 0.10 mm and 1.00 mm are applied. These thresholds can be regarded as low values for Slovene conditions where annual precipitation ranges from around 900 mm to more than 3,000 mm (de Luis et al., 2012). Applying

these thresholds reduces the rE of number of wet time steps to 41%, 24 % and 19 % for ERA5-Land and 8 %, -4 % and -3 % for COSMO-REA6, respectively. For ERA5-Land, all studied precipitation characteristics in Table 4 improved or kept a similar value. For COSMO-REA6 it is similar, except for the wet spell amount which shows a slight decrease from -20 % (for threshold of 0.01 mm) to -23 % (for 1.0 mm). So, in general, the rainfall process of higher rainfall intensities is better represented by the PRP than the overall rainfall processes.

However, there are differences between the studied PRP (Figure 5). While both underestimate the wet spell amount similarly, ERA5-Land overestimates the wet spell duration and leads to an underestimation of the rainfall intensities. COSMO-REA6 underestimates the wet spell duration slightly, which leads to a slighter underestimation of the rainfall intensities. It should be noted that this comparison is only valid for a threshold of 1.0 mm due to the high number of small rainfall intensities for ERA5-Land, as mentioned before.

As for the rainfall intensities, also the extreme values are underestimated (Figure 6). ERA5-Land leads to a stronger underestimation than COSMO-REA6. The medians of rE for return periods of $Tn$ = {1, 2, 5, 10, 20, 50 years} for ERA5-Land ($rE_{median}$ = {-34, -34, -35, -35, -35, -35 %}) and for COSMO-REA6 ($rE_{median}$ = {-19, -19, -19, -19, -19, -19 %}), show relative constant underestimations, on average, with -35 % for ERA5-Land and -19 % for COSMO-REA6, respectively (Figure 6).

Previous studies have shown large variability in case of extreme and short-duration rainfall events in Slovenia (Dolšak et al.,
2016)) that is a consequence of climatic diversities in this region (Vreča et al., 2006).

**Table 4: Mean relative error rE (in %, over all 16 catchments) for daily precipitation characteristics in comparison with ARSO-d in dependence of the applied threshold.**

| Precipitation characteristic | PRP | Threshold [mm/d] | | |
|---|---|---|---|---|
| | | ≥ 0.01 | ≥ 0.10 | ≥ 1.00 |
| Number of wet time steps | ERA5-Land | 41 | 24 | 19 |
| | COSMO-REA6 | 8 | -4 | -3 |
| Total precipitation amount | ERA5-Land | -13 | -13 | -14 |
| | COSMO-REA6 | -21 | -22 | -22 |
| Wet spell duration | ERA5-Land | 93 | 33 | 10 |
| | COSMO-REA6 | 10 | -6 | -5 |
| Wet spell amount | ERA5-Land | 19 | -7 | -20 |
| | COSMO-REA6 | -20 | -23 | -23 |
| Dry spell duration | ERA5-Land | -32 | -21 | -16 |
| | COSMO-REA6 | -9 | 2 | 0 |
| Fraction of dry intervals | ERA5-Land | -50 | -26 | -10 |
| | COSMO-REA6 | -10 | 4 | 2 |
| Average intensity | ERA5-Land | -38 | -30 | -28 |
| | COSMO-REA6 | -27 | -18 | -20 |


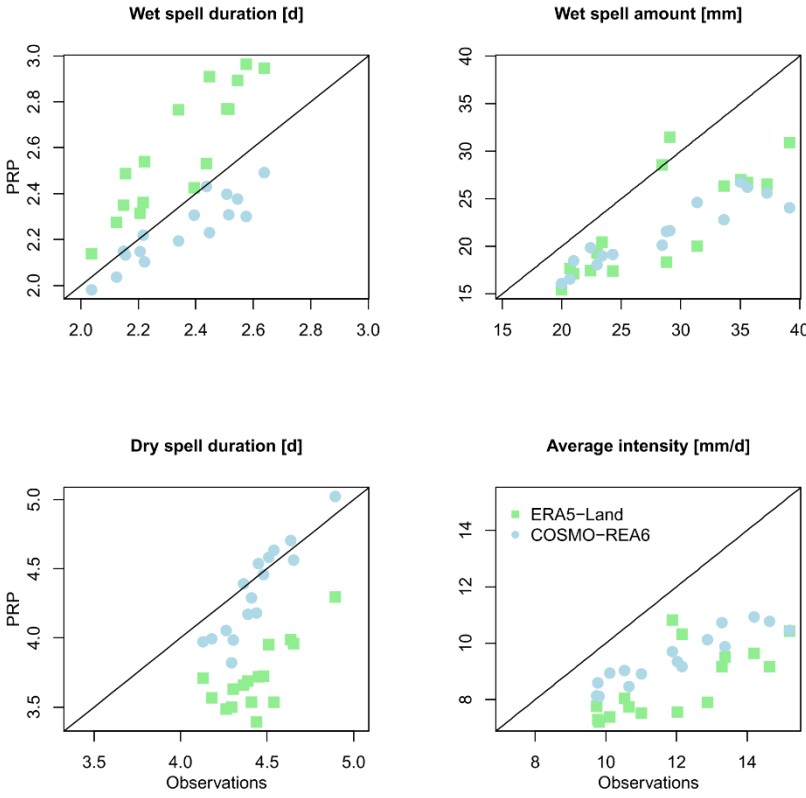

**Figure 5: Comparison of the selected precipitation characteristics of ERA5-Land and COSMO-REA6 in comparison to observations for rainfall intensities >= 1.0 mm.**


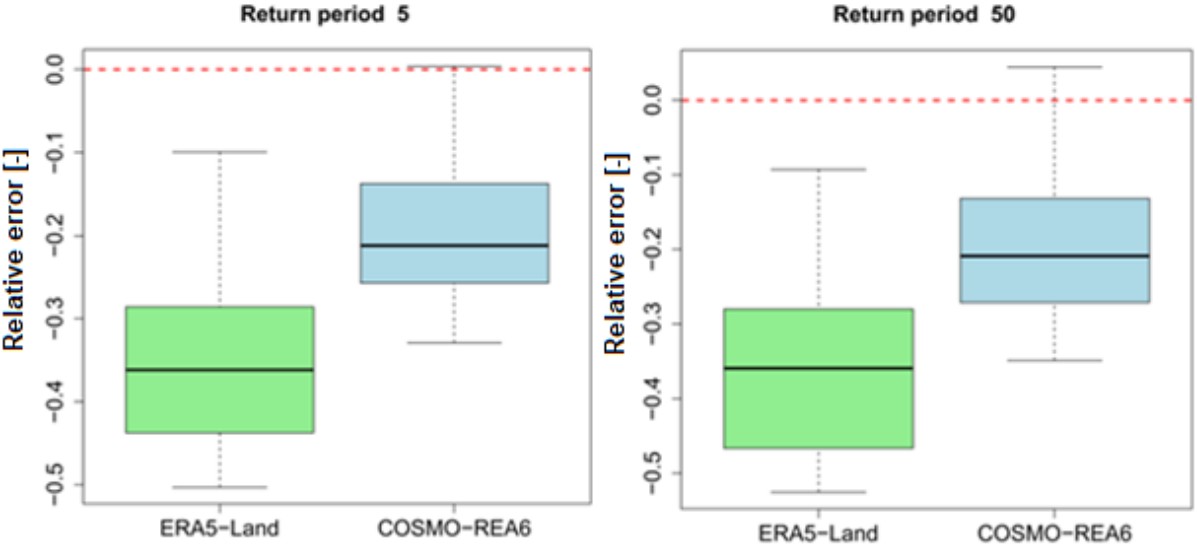

**Figure 6: Deviations of areal rainfall extreme values (5- and 50-year return periods) of ERA5-Land and COSMO-REA6 in comparison to observations over all 16 catchments.**

### 4.2 Validation of air temperature reanalysis products

The validation results for the temperature time series of the reanalysis products are shown in Figure 7. The quality depends strongly on the studied catchment as shown in Figure 7 (top). For Kokra-Kokra catchment, a strong overestimation can be identified from April to November, while for the winter months a quite good fit can be identified. On the other hand, for the Reka-Cerkvenikov mlin (CMlin) catchment, a good fit can be identified for the whole year. In general, the winter months are better represented for all catchments, so the range of the absolute error is smaller than the summer months (Figure 7, bottom left). It should also be noted that the difference between ERA5-Land and COSMO-REA6 is quite small in comparison to the deviations from observations. Therefore, no general conclusion is possible, which reanalysis product represents the air temperature observations better. Additionally, the influence of the catchment elevation on the deviations is shown in Figure 7 (bottom right). For catchments above 1000 m a.s.l. only overestimations can be identified. Similar conclusion was also made in study conducted by Mikoš et al. (2022), that compared ERA5-Land with station based air temperature values when preparing the freez-thaw map of Slovenia. For the majority of the lower located catchments underestimations are identified. However, no clear trend can be identified, neither for ERA5-Land nor COSMO-REA6 (Figure 7).

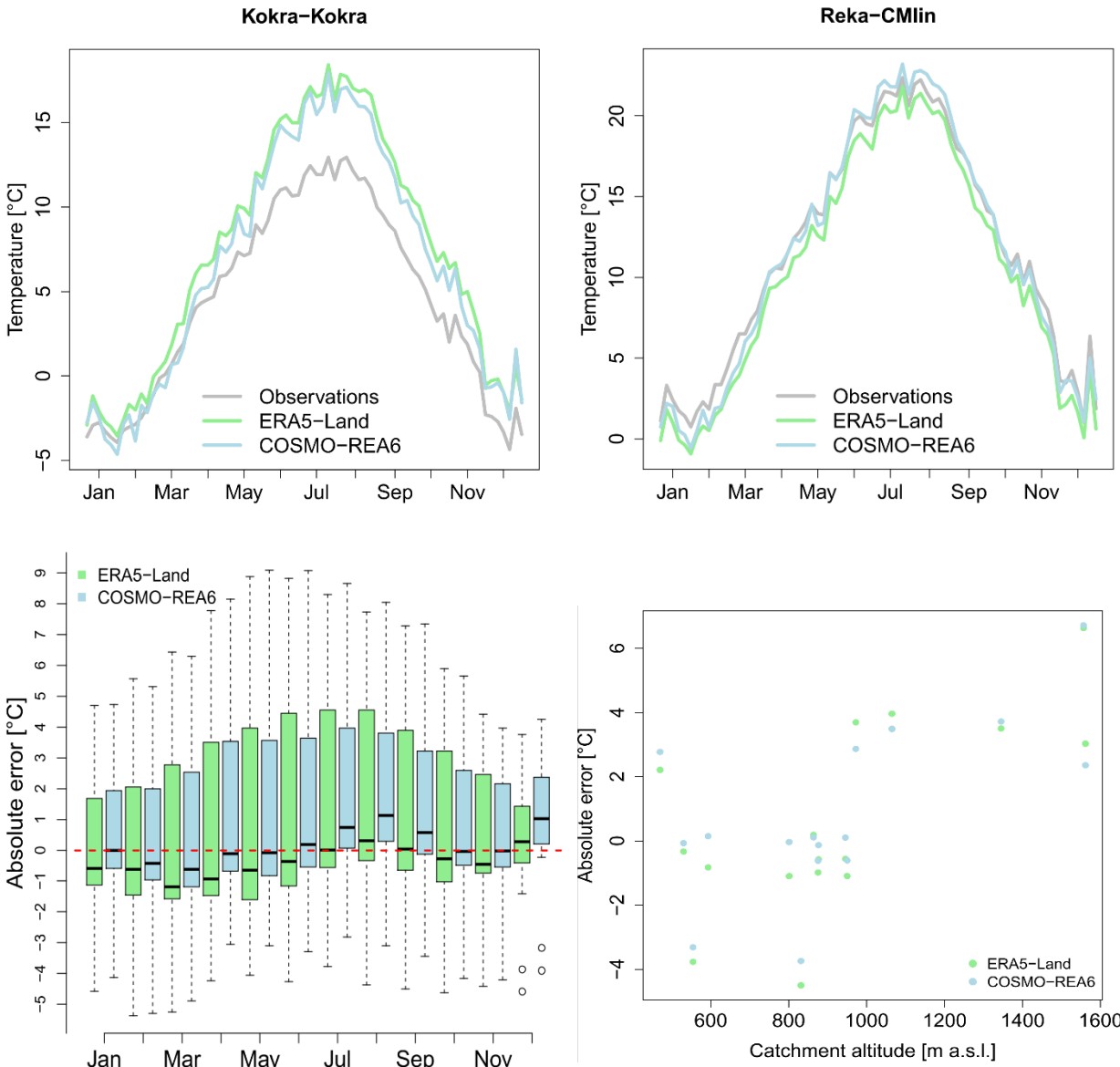

**Figure 7: Top left and right: Median of daily temperature value smoothed via moving window of 5 days length for two selected catchments. Bottom left: Box plots of median monthly values for all considered catchments. Bottom right: Mean of monthly absolute error medians in dependence of catchment elevation.**

**4.3 Validation of reanalysis products using rainfall-runoff modelling**

Figure 8 offers a visualization of the time series for each configuration within the GR4H hydrological model, specifically
applied to the Mislinja-Otiški vrh watershed during the calibration phase.

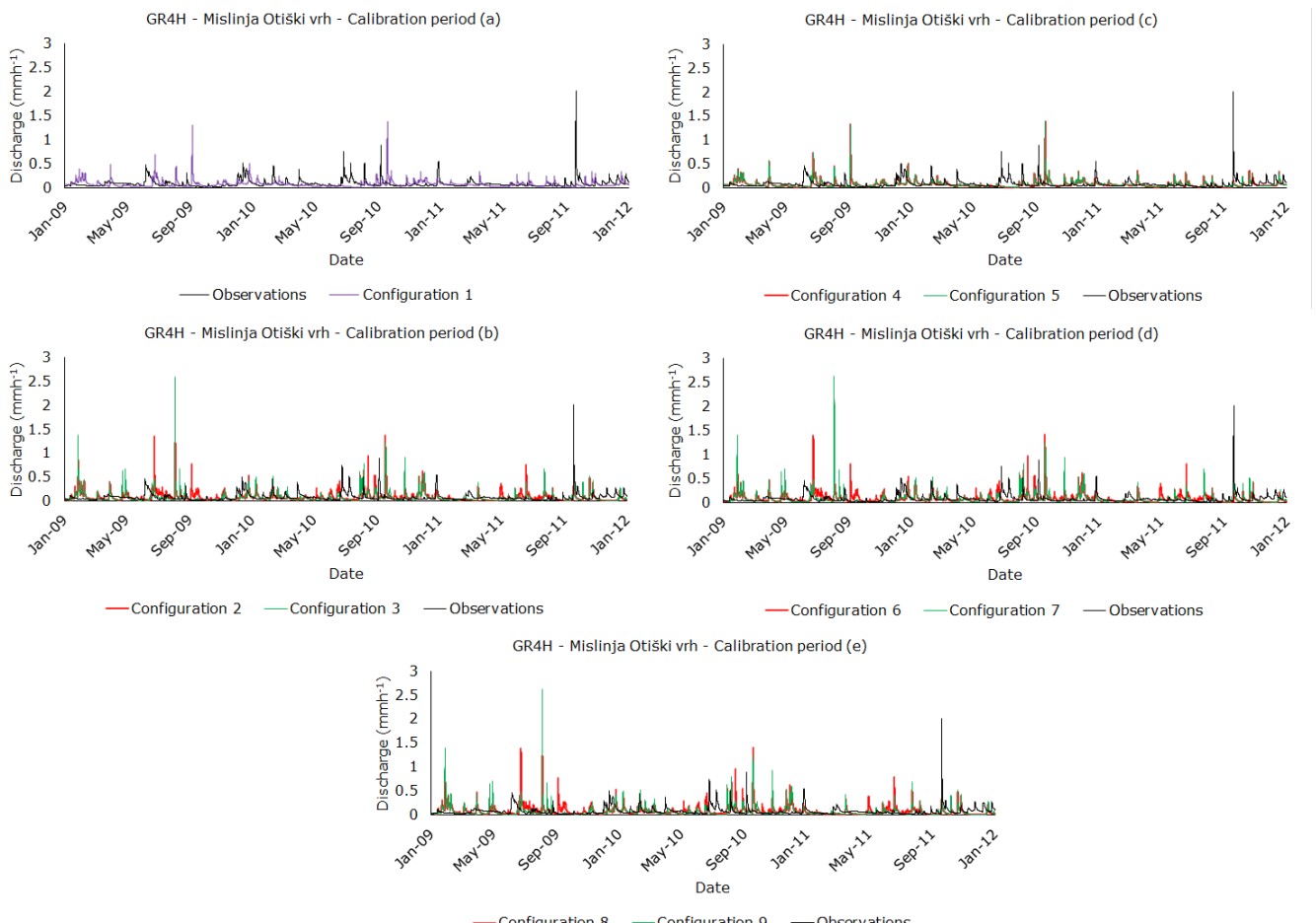

**Figure 8: Time series for the Mislinja-Otiški vrh - GR4H - Calibration period. All Configurations.**

Figure 8 (a) presents the time series of the observation data juxtaposed against configuration 1. The successful calibration of
the model is evident, as corroborated by Figure 9, which occasionally displays overestimations with the phenomenon being
more pronounced in September, averaging a difference of approximately 1 mm per hour.

Figure 8 (b) illustrates the time series generated when employing the ERA5-Land and COSMO-REA6 precipitation reanalysis
products (PRPs) as input variables. When compared with the observations, both PRPs tend to overestimate actual discharge
rates during periods of low (October - June) and high flows (June - September), with this effect being more prominent for the
years 2009 and 2011. Among the two PRPs, COSMO-REA6 exhibits greater overestimation than ERA5-Land. Additionally,

both products fail to adequately capture the observed peak that occurs in September 2011. This is reflected in the KGE values, which range between 0.2 and 0.4 for COSMO-REA6 and 0.4 and 0.6 for ERA5-Land, respectively.

Figure 8 (c) displays the runoff time series when observed precipitation and reanalysis temperature data are used for the calculation of potential evapotranspiration. It becomes evident that the model resembles the behaviour of configuration 1, and configurations 4 and 5 essentially overlap, indicating that temperature is not as significant a parameter as precipitation in the
rainfall-runoff process. Figures (d) and (e) showcase configurations 6, 7, and 8, 9, respectively. In these instances, model performance is hindered, with considerable overestimation of observed values, similar to the case in Figure 8 (b).

Figure 9 illustrates the KGE scores for each configuration (Table 3) as described in section 3.3. Results are presented for the two initial runs, where model variable calibration is performed exclusively for observed precipitation and air temperature. Thirteen (GR4H) and eleven (GR4H Cema Neige) catchments out of the initial selection are successfully calibrated (KGE >
0.6), in spite of the aforementioned distance of the representative station from their centroid. The smaller number of successfully calibrated catchments using model with the snow module can be attributed to possible misapplication of the empirical equation of snow percentage, since it could introduce excess snow quantity that is not applicable to the catchments located in the northern part of the country (i.e., Kokra-Kokra, Radovna-Podhom, Dravinja-Zreče). This is justified as their performance is hindered in both study periods. However, the implementation of the snow component leads to an increase in
performance for the Savinja-Nazarje, Poljanska Sora-Zminec and Selška Sora-Železniki catchments during the validation period. Moreover, the poorer calibration performance in Bolska-Dolenja vas (Dvas) and Savinja-Nazarje catchments could be a result of the location of the weather station. The position of the station during calibration for said catchments is relatively far and may not portray adequately rainfall dynamics (Figure 3). Interestingly, KGE scores for configuration 1 decreases by 0.2 during the validation period for ten and six catchments in the GR4H and GR4H Cema Neige modules, respectively. The smaller
decrease using the model with the snow module validates the assumption that the snow account factor is improving the calibration process by introducing more variables. Similar conclusion was also made by (Lavtar et al., 2020), when conducting rainfall-runoff modelling for several nested catchments within the Sava River basin in Slovenia.

In terms of precipitation reanalysis data, configurations 2 (purely ERA5-Land) and 3 (purely COSMO-REA6) perform similarly across both modules and for both study periods in the Idrijca-Hotešk and Radovna-Podhom catchments. This may
suggest a correlation with their spatial location in the north-western part of the country (Figure 1), since their calibration performance is varying across the study period or the model examined. ERA5-Land is consistently outperforming COSMO-REA6 in Bolska-Dolenja vas (Dvas), Dravinja-Zreče, Lahinja-Gradac and Savinja-Nazarje catchments. No correlations can be made to their catchment characteristics (Figure 10), since they follow different discharge regimes and characteristics (e.g., percentage of forest area, elevation, slope or catchment area) are significantly different. It is assumed that the coarse spatial
resolution of ERA5-Land hinders to capture rainfall dynamics in case of Reka-Cerkvenikov mlin (CMlin) and Rižana-Kubed catchments (KGE values out of bounds), which follow a Mediterranean pluvial regime (Frantar et al., 2008; Frantar and Hrvatin, 2008) and are located in the extreme south-western part of the country, even though they are well calibrated during the rainfall-runoff process (Figure 9). COSMO- REA6 tends to produce negative KGE values for a selection of catchments

(Figure 9). It consistently fails to reproduce streamflow in the Dravinja-Zreče and Hudinja-Škofja vas (SVas) catchments, even during the calibration period, where their performance under configuration 1 is relatively good (Figure 9). Since these catchments are geographically adjacent to each other, this could imply a faulty forcing for COSMO-REA6 in the specific area. Furthermore, performance is examined against catchment characteristics (Figure 10). Figure 10 displays PRP performance (configurations 2, 3) against catchment area, median elevation, slope and percentage of forest area, for the GR4H module during the calibration period. No significant relationships to tested variables can be identified. The assumption is that a larger drainage area incorporates more gridded information from the reanalysis product, and therefore should be introducing less bias. This may be the case for Voglanja-Črnolica catchment (i.e., GR4H – Calibration period). However, COSMO-REA6 has an average KGE value for Kolpa-Petrina, whose initial calibration was successful, and its area coverage is the greatest amongst the selection.

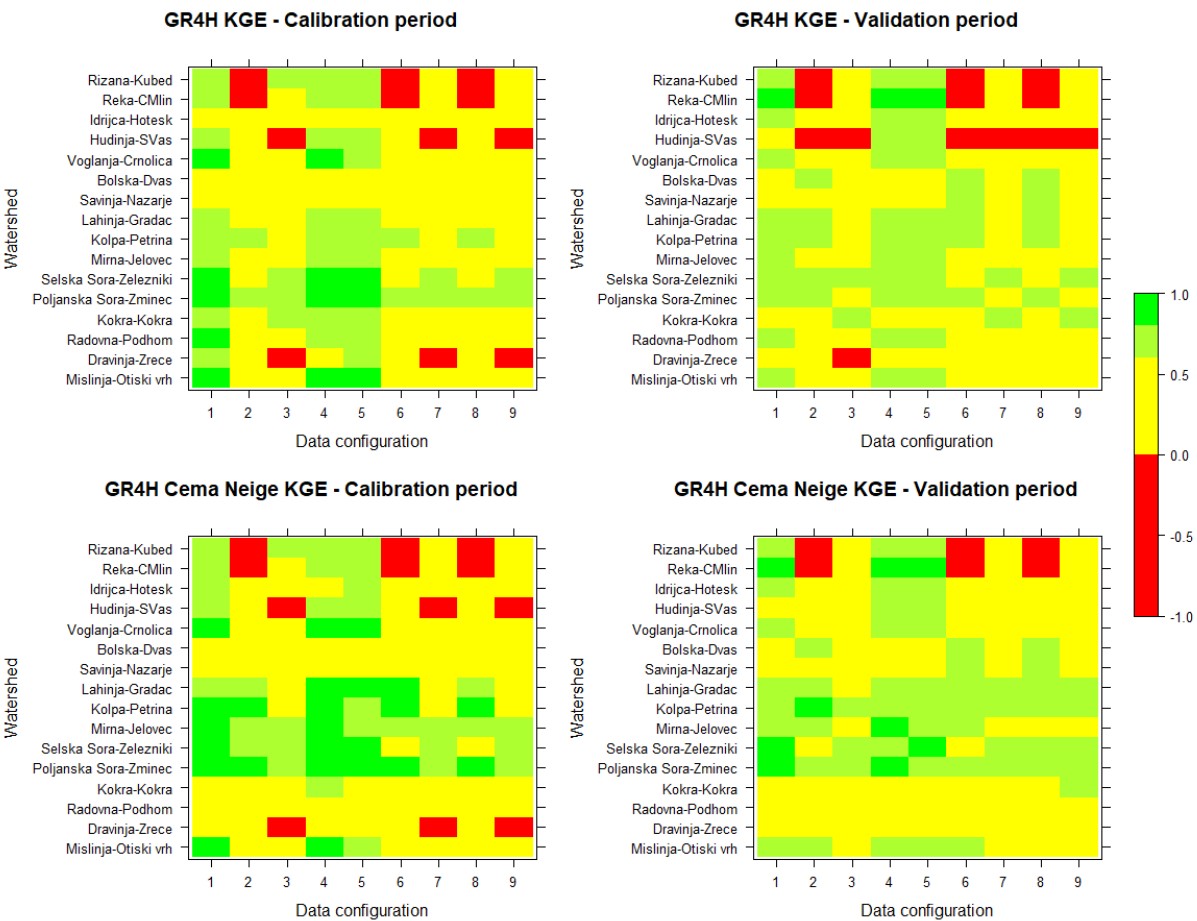

**Figure 9: KGE scores for rainfall-runoff simulations per configuration for each catchment (initial model variable calibration). KGE values are distributed in five classes. White cells indicate negative values.**

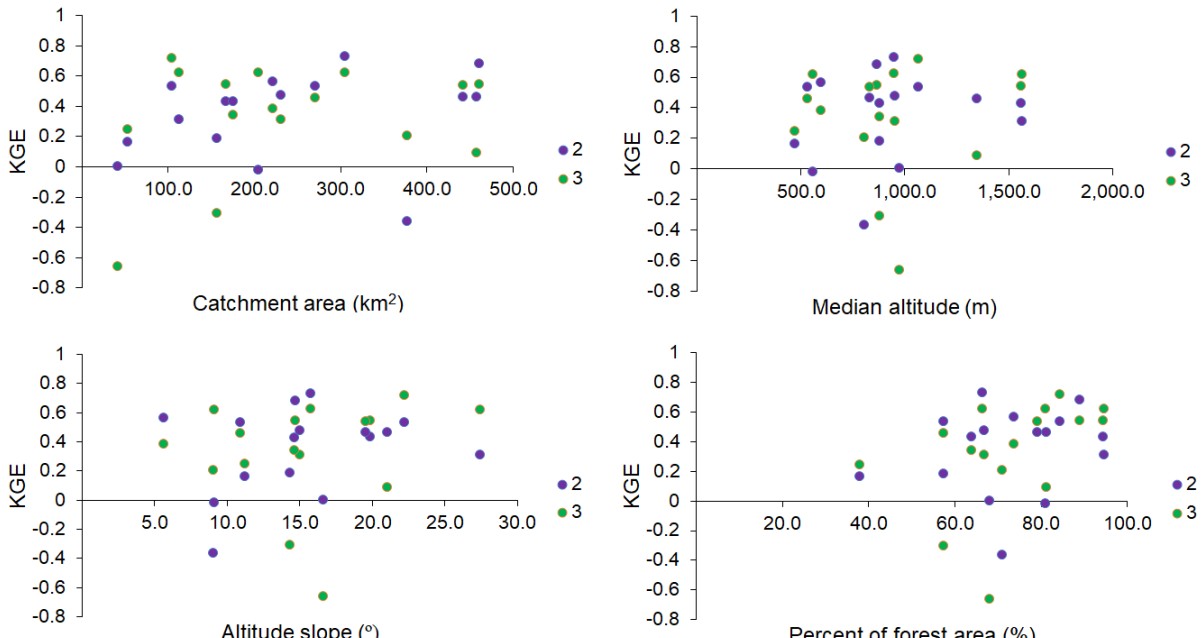

**Figure 10: KGE scores against catchment characteristics for configurations 2 and 3 (Table 3).**

Furthermore, configurations 4 and 5 do not display significant deviations from the initial performance under configuration 1 (Table 3). The inter-comparison between the two reanalyses shows that they perform rather similarly in most catchments, and KGE values are heavily dependent on initial model calibration (Figure 11).

Minor differences are observed within the snow module, where temperature is used as a direct input during the rainfall-runoff process. However, due to the similarity of the temperature data from both reanalysis products KGE values remain close to each other. This aligns with the findings of Bezak et al. (2020) who applied GR4J, GR6J and CemaNeige GR6J hydrological models for some catchments in Slovenia and argued that air temperature as input data has minor influence on rainfall-runoff modelling results in comparison to precipitation data.

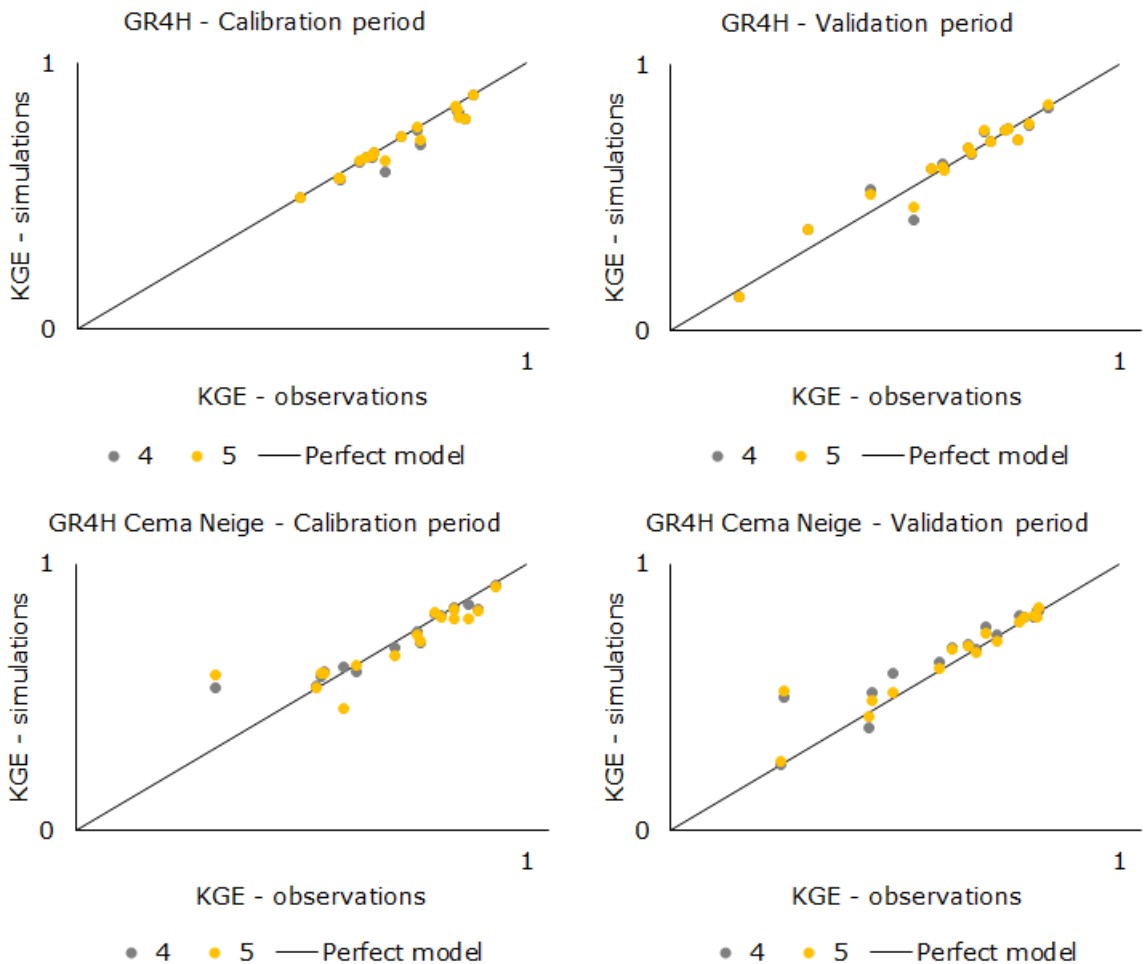

**Figure 11: KGE values of reanalysis temperature against observed temperature for configurations 4 and 5.**

Figure 12 displays KGE scores for the GR4H and GR4H Cema Neige models, where model parameter calibration (e.g., *X1: production store maximum capacity*) is conducted on every configuration by implementing the Michel calibration algorithm. By recalibrating the model parameters whenever a reanalysis input is inserted, regardless of the configuration, performance is significantly improved. Re-initiating the calibration process modifies the ingrained parameters: if e.g., the selected PRP is overestimating the amount of rainfall in the catchment, the storage capacity *X1* is increased in order to account for the excessive quantity. The exchange coefficient *X2* is decreased, therefore less water is imported in the routing storage. The capacity of the routing storage is also increased (*X3*), therefore less runoff water is derived from UH1. In the GR4H module - calibration period, ERA5-Land maintains KGE values above the 0.6 benchmark across the selection (configurations 2, 6, 8). Notable exception is apparent for catchments located in the north eastern part of the country. Similar patterns are followed by COSMO-

REA6. Additionally, an intercomparison between runs with both reanalysis products used as an input was carried out, and performance is identical within the GR4H module. In the snow module, ERA5-Land precipitation offers a slight better performance when coupled with ERA5-Land temperature in a minority of catchments (except for Radovna-Podhom – GR4H Cema Neige – calibration period), however most results are consistent. The same applies for the COSMO-REA6. Moreover, ad-hoc calibration is leading to partial performance improvement under configurations 4 and 5 for the GR4H Cema Neige module (Figure 12). Ad-hoc calibration refers to the implementation of the Mitchel algorithm with each new insertion of reanalysis within the model; that is, instead of keeping the model calibrated using observations, its ingrained parameters are re-optimized in order for the reanalysis to converge closer to discharge observations. The effect is more pronounced during the calibration period. In the module considering the snow factor, ad-hoc recalibration yields better results. However, this observation is not consistent across all catchments (e.g., Reka-Cerkvenikov mlin (CMlin) – validation period – COSMO-REA6). When using both reanalysis products in the rainfall-runoff simulations (configurations 6-9), ad-hoc recalibration proves to be effective, and model variables are able to correct faulty forcings before the simulation process. Exceptions are present for COSMO-REA6 in the GR4H module during the validation period in the Bolska-Dolenja vas (Dvas) and Savinja-Nazarje catchments, where the initial calibration is not adequate. Overall, no configuration is deemed favorable, since each is displaying subpar performance on average for four catchments during both study periods.

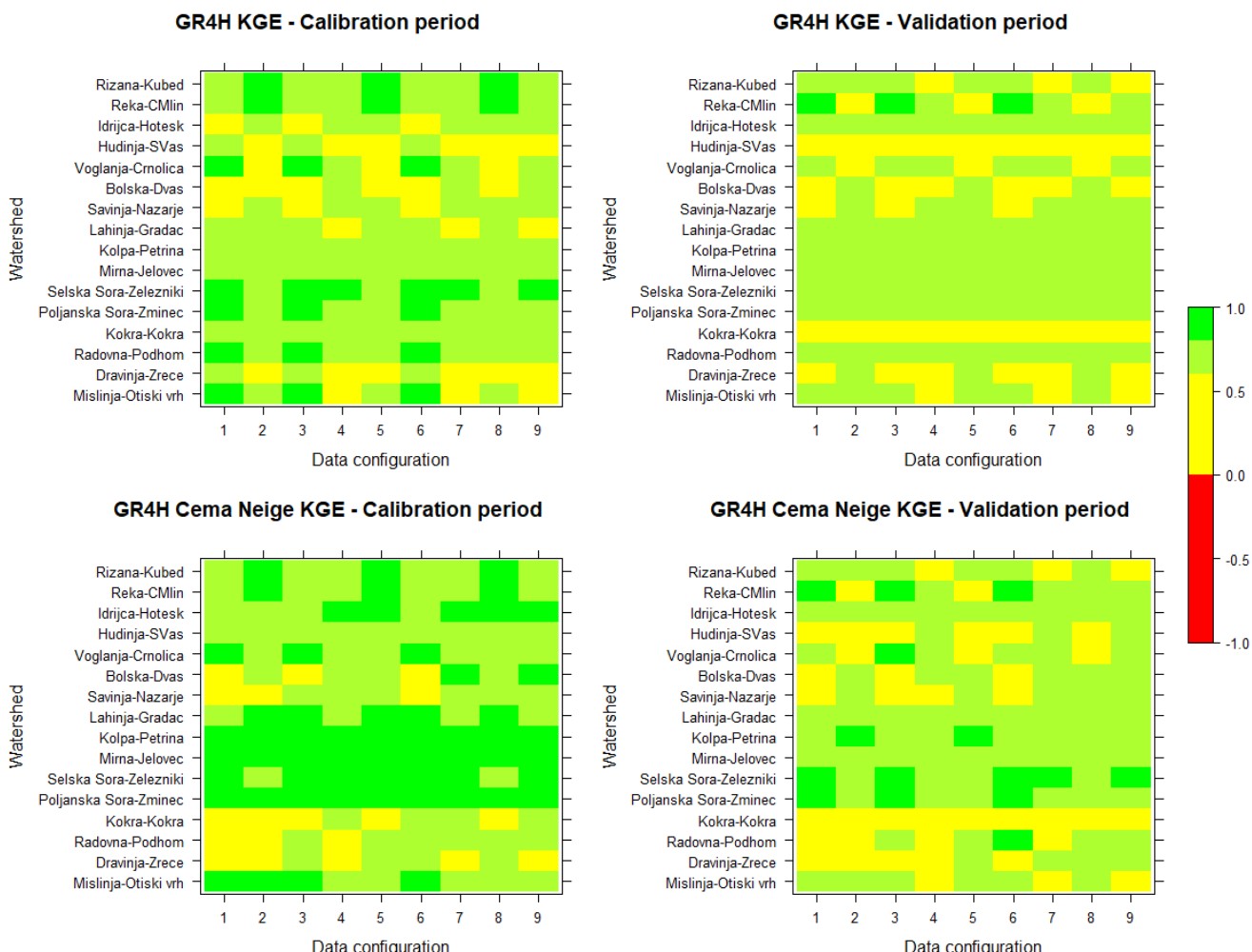

**Figure 12: KGE scores for rainfall-runoff simulations per configuration for each catchment (ad-hoc calibration). KGE values are distributed in five classes.**

An additional investigation regarding catchment location is shown in Figure 13, where scatterplots illustrate performance in relation to catchment location. No substantial trends are observable in relation to latitude. However, the linear regression conducted for KGE against longitude illustrates that horizontal coordinates account for KGE variation by 76%. ERA5-Land forcing's show more bias towards the east, which could be related to the moisture sources of precipitation in Slovenia where almost half of precipitation originates from the central and western Mediterranean (Krklec et al., 2018).

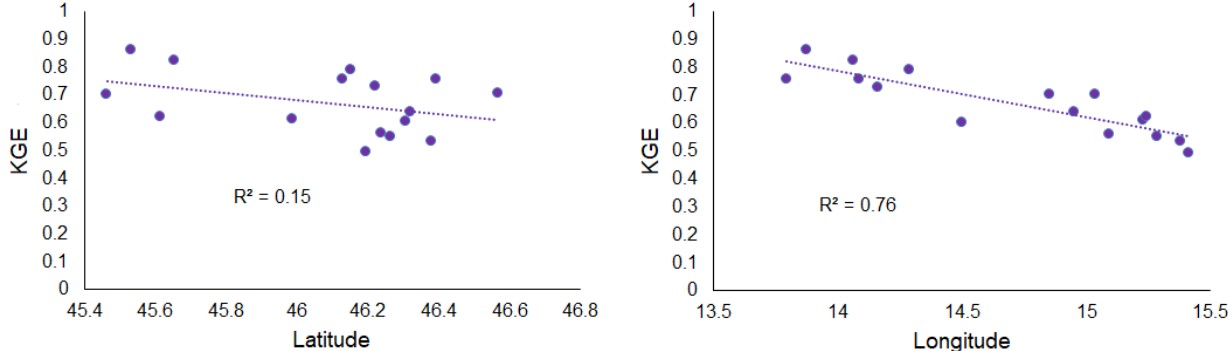

**Figure 13: KGE scores (configuration 2 – GR4H – Calibration period), relative to longitude and latitude.**


### 4.4 Study limitations and lessons learned

It should be noted that there are several limitations related to the conducted study, which should be mentioned specifically:

For the validation of the reanalysis products, different observation data sets were having a positive or negative impact on the results. Though, one motivation for reanalysis products is to generate data for (partly) unobserved regions, which hinders their

validation for these areas. In this study, the most representative data sets from the authors perspective were chosen for the validation. Also, the spatial and temporal resolutions differ among the data sets used. If possible, the authors unified the resolutions for equitable comparisons (e.g., aggregation of time series to daily values for temporal unification; or estimation of areal time series based on catchment boundaries rather than on underlying rasters with different resolutions).

For the rainfall-runoff simulations, station-based time series were used as input with no spatial interpolation. A spatial

interpolation would have led to a more equitable comparison, since the reanalysis data includes spatial information as well. Nevertheless, this was not possible due to the limited number of nearby stations with hourly resolution. It can be assumed that simulated runoff results would have been more similar between observations and reanalysis data, as indicated by the strong improvement from the re-calibration of the GR4H and GR4H Cema Neige models for each data set.

Also, GR4H and GR4H Cema Neige are both lumped models, so any spatial benefit resulting from the PRP is of little added,

value compared to a spatially distributed model. Since spatial effects increase with catchment size it would be interesting to study the impact of the model choice (lumped, semi-distributed or fully distributed) in relation to catchment sizes, and if there is an impact identifiable for the studied catchments ($<500$ km$^2$). However, due to the aforementioned data scarcity issues, the authors decided to apply the lumped model often used in this region for comparisons. This study has offered useful insights into the hydrological response of the catchments studied in relation to the selected PRPs. The next step in this field of research

would be to evaluate the use of satellite and other means of remote sensing data to obtain a thorough overview of the available

options of handling missing weather station data in Slovenia, Central Europe. Additionally, conducting an uncertainty analysis could potentially offer valuable insights by quantifying uncertainty in the PRPs and identifying their effects during model propagation. Nonetheless, it falls beyond the scope of the current research, and the investigation of uncertainty analysis and its implications on the model's outcomes are left for future studies to explore.


## 5 Conclusions

For 16 catchments in Slovenia, the precipitation reanalysis products COSMO-REA6 and ERA5-Land are validated as a possible input for rainfall-runoff modelling in data scarce regions. The validation of the areal rainfall time series leads to the following conclusions:

- ERA5-Land has a high fraction of wet time steps with very small rainfall intensities, which should be excluded before rainfall characteristics are validated.

- COSMO-REA6 leads to a better representation of number of wet time steps, average intensity, wet and dry spell duration. ERA5-Land leads to a better representation of the total rainfall amount and wet spell amount.

- Both, COSMO-REA6 and ERA5-Land, underestimate the rainfall extreme values. For return periods Tn={1, 2, 5, 10,
20, 50 years} ERA5-Land shows underestimations of -34 %, whilst COSMO-REA6 of -19 %.

The conclusions from the comparison of air temperature data (hourly time step, 5 years of data) are:

- ERA5-Land and COSMO-REA6 show similar deviations from observations over all catchments, with smaller deviations during winter months.

- For catchment elevations >1000 m a.s.l. overestimations are identified for both reanalysis products. For lower located
catchments deviations are smaller, but the pattern is less clear.

A generalisation of these conclusions is limited due to the regional differences of rainfall processes and the ability of the reanalysis models to represent them. However, the more similar areas of interest are in terms of hydro-climatology, the more likely similar findings can be expected.

Additionally, multiple rainfall-runoff simulations were performed at an hourly time step. Based on the conducted simulations,
the following conclusions can be made:

- The selected air temperature dataset has a smaller impact on the rainfall-runoff modelling performance than precipitation. Hence, temperature reanalysis data offers a viable option for rainfall-runoff modelling at the hourly timestep, providing no significant differences with observations in terms of performance.

- When using PRP, the GR4H Cema Neige yields in general better results compared to the GR4H model, especially
during the calibration period, which can be explained with two additional parameters that Cema Neige uses. This is even more significant for the Alpine catchments with pronounced snow cover during winter.

- ERA5-Land shows slightly increasing bias towards the eastern direction (catchment location) which can be related to the origin of main moisture sources in Slovenia.

- Non-bias corrected ERA5-Land and COSMO-REA6 values produce slightly better results in case of large catchments compared to smaller ones. In most cases the rainfall-runoff modelling performance using ERA5-Land is slightly better compared to the COSMO-REA6.

ERA5-Land and COSMO-REA6 can be used as input data for hourly rainfall-runoff models and provide an alternative data source for a significant domain of Central Europe, characterised as a transitional zone between Mediterranean and Continental climate. If a re-calibration is carried out, the runoff simulations with PRP show similar performance measures while at the same time offering temporally and spatially continuous availability over many decades. However, their performance is varying and it is not significantly related to catchment characteristics; at least not the ones tested within this study. More research is needed to test the performance on a larger number of catchments, in addition to implementing a bias-correction method for the PRP to further investigate their potential application in rainfall-runoff studies.

### Code and data availability

The COSMO-REA6 regional reanalysis is publicly available via DWD's Climate Data Center: https://opendata.dwd.de/climate_environment/REA (DWD/HErZ, 2020). The Global ERA5-Land reanalysis is publicly available via the Copernicus Climate Data Store (CDS) (Muñoz Sabater 2019). Data from ARSO can be obtained upon request (gp.arso@gov.si). Further data and code used for calculations can be obtained from the first and corresponding author upon request.

### Author contribution

All authors developed the concepts of the manuscript. P.N. pre-processed the reanalysis products, M.J.A. conducted rainfall-runoff calculations and analysed the simulation results. H.M.-T. validated the rainfall and temperature time series. M.J.A. and H.M.-T. wrote the first draft. H.M.-T., P.N., N.B., and M.Š. edited and improved the manuscript and figures.

### Competing interests

Authors declare no conflict of interest.

### Acknowledgements

We would like to acknowledge the Slovenian Environment Agency (ARSO) for data provision. We would also like to thank the valuable comments provided by anonymous reviewers and editorial board members that greatly improved the quality of this manuscript.

### Funding

N. Bezak and M. Šraj would like to acknowledge the support of the Slovenian Research Agency (ARRS) through grants P2-0180, V2-2137, N2-0313, and J6-4628 and support from the UNESCO Chair on Water-related Disaster Risk Reduction. The results are also part of the bilateral project between Slovenia and Germany "Validation of precipitation reanalysis products for rainfall-runoff modelling in Slovenia (PRE-PROMISE)", funded by the German Federal Ministry of Education and Research (BMBF).

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
