# Peer review of "Validation of precipitation reanalysis products for rainfall-runoff modelling in Slovenia"

_EGUsphere, 2022_

## Referee Comment (RC2)

**Validation of precipitation reanalysis products for rainfall runoff modelling in Slovenia**

The authors present a study evaluating two reanalysis products against a gridded precipitation dataset that was constructed based on ground observational data from errors in input sources and errors propagated into hydrologic simulation across 16 catchments in Slovenia. The paper is within the scope of EGUsphere, however, major revisions must be made before considering publication. Below lists my major concerns regarding the scope of this study, methodology, and results presentation.

Major comments:

1. The scope or motivation of this study is not well articulated. Apparently, reanalysis datasets are not used to solve precipitation, but rather other meteorological variables. Precipitation variable is a by-product. Most of the reanalysis datasets do not directly assimilate precipitation into model chain (Although ERA5 and MERRA2 did assimilate precipitation). For hydrologic utilities - like floods the authors mentioned, a reanalysis dataset is not qualified and should not be a supplement for real observations unless no observations available. In summary, the authors should provide a strong argument prior to their analysis - why use reanalysis for hydrologic modelling since observational datasets exist.

2. In the introduction, I am not convinced by the facts why the authors chose to evaluate reanalysis data. In line 61- 63, the authors stated that " mostly data gathered by means of remote sensing technology. ". First, I would like the authors to state why they are not considering other satellite derived products which have been proven to perform better than reanalysis. I am concerned the scope of this study. See my first comment.

3 The authors site a range of literature on evaluation of reanalysis in Lines 52- 93. However, I don't see a summary of what the reviewed studies did not do, that is compensated by the current study and why reanalysis for Slovenian catchments.  The authors should make this clear in the introduction.

3. Why is the ARSO-d data considered as the benchmark? There are other observational precipitation datasets available such as IMERG, MSWEP and GSMaP. The authors should provide more information to claim that ARSO-d data is more accurate than reanalysis products. More details of the on this data should be provided, such as data quality, and the data source, validation etc.

4. How is calibration applied to different forcing?

5. Include the location of the catchments in Table 1 (Lat and Lon), catchment characteristics relevant in the formation and dominant hydrological processes

6. In line 250 – 254, the authors state "The simulation period is split-sampled into….."but do not justify the use of Split Sample method in model calibration. Refer to Arsenault et. al., 2018. The hazards of split-sample validation in hydrological model calibration. *Journal of hydrology*, *566*, pp.346-362.

7.  Line 268 – 273 Please check the font and line spacing

8. There are many recent references focusing on evaluation studies and application, you may find it helpful.
Wanzala, M. A., Ficchi, A., Cloke, H. L., Stephens, E. M., Badjana, H. M., & Lavers, D. A. (2022). Assessment of global reanalysis precipitation for hydrological modelling in data-scarce regions: A case study of Kenya. *Journal of Hydrology: Regional Studies*, *41*, 101105.

Tang, G., Clark, M. P., Papalexiou, S. M., Ma, Z., & Hong, Y. (2020). Have satellite precipitation products improved over last two decades? A comprehensive comparison of GPM IMERG with nine satellite and reanalysis datasets. *Remote sensing of environment*, *240*, 111697.

---

## Author Response (AR1)

Reply to reviewers for the manuscript:

**Validation of precipitation reanalysis products for rainfall-runoff modelling in Slovenia**

by Marcos Julien Alexopoulos, Hannes Müller-Thomy, Patrick Nistahl, Mojca Šraj, and Nejc Bezak

The authors thank both reviewers for their constructive comments on the manuscript, which will help to improve it. Please find below detailed reply to all comments in blue. Also, where appropriate, we suggest a possible modification of the manuscript to incorporate our reply (in green).

**Reviewer #1:**

I have read the manuscript entitled " Validation of precipitation reanalysis products for rainfall-runoff modeling in Slovenia", which is a comparative analysis of two reanalysis precipitation datasets, ERA5-Land and COSMO-ERA6, to runoff simulation over different regions of Slovenia

***the manuscript is relatively interesting and well written.*** I consider that it is suitable for publication in the EGUsphere journal, although, I would like to give some comments to the authors for the general improvement of the manuscript.

We thank reviewer #1 for the acknowledgement of the manuscript suitability to HESS, for her/his time and efforts and the resulting comments.

1- Please use more recent papers that are relevant to your topic. Below you can find some new studies:

https://link.springer.com/article/10.1007/s11269-022-03081-9

https://www.tandfonline.com/doi/full/10.1080/02626667.2019.1691217

We updated the introduction with more recent manuscripts as follows:

Line 63:

Several studies on inter-comparisons between various reanalysis products exist in order to identify their suitability for a particular region (Koohi et al., 2022).

Line 99:

Ghodichore et al. (2018) addressed the applicability of the APHRODITE (Yatagai et al., 2009, 2012), ERA-Interim, PERSIANN (Hsu et al., 1997; Sorooshian et al., 2000) and TMPA-RT (Systems et al., 2000; Gebere et al., 2015) reanalysis products over the Sefidrood catchment in Iran, at the daily and monthly timestep. At the latter, all products perform in a similar fashion, with APHRODITE

performance slightly exceeding the rest of the selection. When increasing temporal resolution, APHRODITE and ERA-Interim are better able to capture rainfall station measurements. Feng et al. (2021) evaluated precipitation reanalysis in the United States. Their comparative analysis shows mixed results: reanalysis is unable to capture rainfall dynamics in the northern part, whilst results are adequate in the rest of the study area. Another study using SWAT assessed the performance of 14 remote sensing products over a macro-scale watershed in Pakistan. Another study using SWAT assessed the performance of 14 remote sensing products over a macro-scale watershed in Pakistan. Amongst 14 satellite-, gauge- and reanalysis precipitation, APHRODITE and JRA-55 are the most adequate in capturing rainfall dynamics at the daily scale (Saddique et al., 2022).

2- Improve the quality of all figures.  For example, in Fig1, 2, texts and numbers are blurred and need to improve.

We would like to thank the Reviewer #1 for this comment. We have seen that the figure quality in the submitted manuscript is low, which is caused mainly by the submission of the manuscript as a single file containing all figures. Each figure itself is of high quality and will be submitted as a single-file for the final manuscript composition. However, we also changed some of the figures (same content, but higher resolution/different format): Fig. 1, Fig. 2, Fig. 4.

3- What are your criteria for mesoscale, large-scale, and microscale categories?

Indeed, this definition depends on the field of hydrologic application. For rainfall-runoff modelling, mesoscale typically refers to catchments with areas ranging from few $10^1$ km² to several $100^2$ km². Of course, catchments with smaller areas are considered as micro-scale, with larger areas as macro-scale. So, the studied catchments all belong to the mesoscale as mentioned in the manuscript. We added this information in the abstract for clarification: (41-460 km²)

In the manuscript it's visible from Table 1: Catchment characteristics.

4- Some of the studied catchments have an Area value lower than 100 km2, While ERA5-Land cells are higher than 100 km2. How did you deal with this issue? I think using this dataset for small catchments is not suitable. However, Still I don't understand your criteria for mesoscale. these catchments fall in the microscale category.

Thanks for this comment. ERA5-Land has a raster width of ~9 km, so the area covered by one raster cell is 81 km² (for REA6 it's 36 km²). To account also for raster cells that only partially cover the studied catchments, the calculation of areal precipitation and temperature from the reanalysis data includes the total overlapping area of each relevant raster cell. The reviewer points out, that the spatial resolution is too coarse to use it as input directly for rainfall-runoff simulations. Indeed, testing this hypothesis is one part of the manuscript. Our findings suggest, that with the applied lumped rainfall-runoff model and especially after recalibration the reanalysis data leads to similar good simulation results as observations (Fig. 12, L482-485).

5- The values of numbers in Table 3 are not clear and I can't check the dataset combinations.

We would like to thank the Reviewer #1 for pointing to this. Table 3 is essential for the understanding of all follow-up results, since it provides the composition of precipitation (circle)

and temperature (box) data from either observations, ERA5-Land and COSMO-REA6. The table is not expected to contain values, the symbols (box, circle) indicate only which data type (temperature, precipitation) from which data set (observations, ERA5-Land, COSMO-REA6) is used for which combination (1-9). We added this information in the revised manuscript to avoid possible confusions.

We added at L288:

The purpose of the table is to indicate the data type (temperature, precipitation) and data set (observations, ERA5-Land, COSMO-REA6) used for each combination (1 – 9), rather than to present specific values. Therefore, the symbols (box, circle) are used to represent the data sources instead of actual values.

6- The authors used the GR4H model for runoff simulation. Based on the model structure (Fig 4), this model doesn't consider the snow component through the modeling. How do you explain this issue for snowy catchments of Slovenia?

Reviewer #1 points out the missing snow routine in GR4H as potential error source for rainfall-runoff modelling of snowy catchments. We agree on this potential issue and thus we compared the simulation results from both the GR4H and the GR4H CemaNeige, model as well. The CemaNeige is a snow module (L240-249). Especially for alpine catchments, GR4H Cema Neige yields better results than GR4H (L472-476).

We added this information at L398:

However, the implementation of the snow component leads to an increase in performance for the Savinja-Nazarje, Poljanska Sora-Zminec and Selška Sora-Železniki catchments during the validation period.

7- For precipitation comparison, which approach is used in this study? Point-scale or Grid-scale approach? However, Based on fig1 and fig3 the density of your ground-gauge observations is not so good.

We agree with reviewer #1 that the results from point- and grid-based comparisons can differ significantly. Due to the different areas covered with station-based observations and reanalysis data we did not carry out any comparisons based on station data (L179-185). An interpolation on hourly level would have been no solution due to the low network density as reviewer #1 pointed out.

Hence, the comparison was carried out on daily values of areal rainfall based on an interpolated raster data set from the Slovenian Environment Agency (ARSO). This is explained in L186-190.

8- It is recommended to plot stream flow time series for a better understanding of model performance for simulating peak and low flows.

Thanks for this suggestion. We agree that the manuscript benefits from visual inspections of the simulated runoff time series. We have added Fig. 8 in the revised manuscript. We added a figure at L369 with additional text at L377.

[Figure]

**Figure 1: Time series for the Mislinja-Otiški vrh - GR4H - Calibration period. All Configurations.**

Figure 1 (a) presents the time series of the observation data juxtaposed against configuration 1. The successful calibration of the model is evident, as corroborated by **Fehler! Verweisquelle konnte nicht gefunden werden.**, which occasionally displays overestimations with the phenomenon being more pronounced in September, averaging a difference of approximately 1 mm per hour.

Figure 1 (b) illustrates the time series generated when employing the ERA5-Land and COSMO-REA6 precipitation reanalysis products (PRPs) as input variables. When compared with the observations, both PRPs tend to overestimate actual discharge rates during periods of low (October - June) and high flows (June - September), with this effect being more prominent for the years 2009 and 2011. Among the two PRPs, COSMO-REA6 exhibits greater overestimation than ERA5-Land. Additionally, both products fail to adequately capture the observed peak that occurs in September 2011. This is reflected in the KGE values, which range between 0.2 and 0.4 for COSMO-REA6 and 0.4 and 0.6 for ERA5-Land, respectively.

Figure 1 (c) displays the runoff time series when observed precipitation and reanalysis temperature data are used for the calculation of potential evapotranspiration. It becomes evident that the model resembles the behaviour of configuration 1, and configurations 4 and 5 essentially overlap, indicating that temperature is not as significant a parameter as precipitation in the rainfall-runoff process. Figures (d) and (e) showcase configurations 6, 7, and 8, 9, respectively. In these instances, model performance is hindered, with considerable overestimation of observed values, similar to the case in

Figure 1 (b).

9- For the estimation of ET, which formulas and datasets are used for running the GR4H model? Please clarify this issue in the revised paper.

9. For the estimation of ET, the method of Oudin et al. (2005) is applied (L221). This method is based on 27 potential evapotranspiration formulas. Maybe this link is useful as well: https://webgr.inrae.fr/en/models/evapotranspiration-model/ .

We added the following explanation in the manuscript at L249:

More details about the formula can be found in https://webgr.inrae.fr/en/models/evapotranspiration-model/, where a workbook template is provided for the estimation of $E$ in Excel spreadsheet format.

10- Why didn't the authors consider the uncertainty analysis for using the GR4H model?

10. Thanks for this useful comment. While uncertainty analysis is certainly an important aspect of hydrological modelling, it is (if done properly) a high-complex process that require additional data and resources. Given the scope of our study, we chose to focus on evaluating the performance of reanalysis without bias adjustment using the GR4H and GR4H CemaNeige hydrological models for streamflow simulation. We think a comprehensive uncertainty analysis is beyond the scope of our study.

We address the issue on the manuscript at L528:

Additionally, conducting an uncertainty analysis could potentially offer valuable insights by quantifying uncertainty in the PRPs and identifying their effects during model propagation. Nonetheless, it falls beyond the scope of the current research, and the investigation of uncertainty analysis and its implications on the model's outcomes are left for future studies to explore.

**Reviewer #2**

The authors present a study evaluating two reanalysis products against a gridded precipitation dataset that was constructed based on ground observational data from errors in input sources and errors propagated into hydrologic simulation across 16 catchments in Slovenia. The paper is within the scope of EGUsphere, however, major revisions must be made before considering publication. Below lists my major concerns regarding the scope of this study, methodology, and results presentation.

We thank Reviewer #2 for the acknowledgement of the manuscript suitability for HESS journal, for her/his time and efforts and the resulting comments.

Major comments:

1. The scope or motivation of this study is not well articulated. Apparently, reanalysis datasets are not used to solve precipitation, but rather other meteorological variables. Precipitation variable is a by-product. Most of the reanalysis datasets do not directly assimilate precipitation into model chain (Although ERA5 and MERRA2 did assimilate precipitation). For hydrologic utilities - like floods the authors mentioned, a reanalysis dataset is not qualified and should not be a supplement for real observations unless no observations available. In summary, the authors should provide a strong argument prior to their analysis - why use reanalysis for hydrologic modelling since observational datasets exist.

1. Thanks for this comment. The hydrology community recognizes that the majority of river basins lack adequate data for comprehensive analysis. Insufficient weather station data poses a significant obstacle for conducting hydrological assessments and gaining a full understanding of the underlying processes in a specific region. Reanalysis datasets and EO derived datasets are increasingly becoming popular alternatives for filling this gap. From a historical point of view rainfall was only a by-product in radar and microwave links as well, but both data sets are actively used nowadays for spatial rainfall estimates. However, to use reanalysis data for commercial purposes, proper validation is necessary, as each product is tailored for different regions worldwide. One of the key advantages of reanalysis datasets is their availability of consistent and reliable data over extended periods, particularly in regions with data gaps. These datasets offer a wealth of information on critical meteorological variables such as temperature humidity, wind and pressure, which are crucial inputs for hydrological assessments, depending on the type and detail of the modelling applied. Regarding the 'existence of observations', we mention in the manuscript that the existing observations on hourly resolution in Slovenia have neither the temporal nor spatial coverage of the reanalysis data – so reanalysis data are an essential benefit for any hydrologic modeller.

We added the following sentence at the end of the introduction:

It will be analysed if the good spatial and temporal coverage of the reanalysis data can compensate possible quantitative deviations of rainfall amounts from the observational station network, which comes therefore with less spatial and temporal coverage.

2. In the introduction, I am not convinced by the facts why the authors chose to evaluate reanalysis data. In line 61-63, the authors stated that " mostly data gathered by means of remote sensing technology. ". First, I would like the authors to state why they are not considering other satellite derived products which have been proven to perform better than reanalysis. I am concerned the scope of this study. See my first comment.

2. Thank you for the comment. Indeed, satellite data can provide valuable information for hydrological studies. However, reanalysis data cover longer periods than satellite datasets. For example, ERA5-Land is available since 1950s and is expected to start in 1940s soon (since ERA5 was updated during the review process and starts now in 1940). In addition, there is literature currently in favour of reanalysis products over satellite for certain regions. For example, the study conducted by Zhang et al. (2021) compared the performance of satellite-based and reanalysis precipitation products for hydrological modelling in the Beipanjiang river basin. The four satellite products used in the study are TRMM, GPM, IMERG and CMORPH, while the reanalysis used is ERA5-Land. The study found that ERA5-land outperforms all four satellite products in terms of accuracy and consistency in both wet and dry seasons. Specifically, ERA5-land shows higher KGE and NSE values and lower MAE values compared to the satellite products. Another study compared three precipitation products in the Upper Blue Nile Basin. The study evaluated the performance of the TRMM, CHIRPS and COSMO-REA6 at the three-hourly timestep. The findings show that COSMO-REA6 slightly outperformed the satellite-based precipitation products in the wet season. COSMO-REA6 displays a KGE value of 0.62, whilst TRMM and CHIRPS had a value of 0.57 and 0.58, respectively. The fact that: i) certain reanalysis products can outperform satellite estimates in specific regions; ii) there is a scarcity of findings regarding the reliability of weather station alternatives in the domain of Slovenia; and iii) some of these alternatives are provided at the hourly timestep, provide in our opinion a solid motivation for conducting the present research.

Zhang, Y., Chen, X., He, Y., Zhu, Y., & Liu, S. (2021). Comparison of satellite and reanalysis precipitation data for hydrological modeling in a large karst catchment. Journal of Hydrology, 597, 126166. doi: 10.1016/j.jhydrol.2020.126166

Gebremichael, G.G., Hossain, F., Erickson, T., Gao, H. & Hopson, T. (2017). Evaluation of the performance of precipitation products for hydrological modeling over the Upper Blue Nile Basin. Journal of Hydrology, 550, 406-418. doi: 10.1016/j.jhydrol.2017.05.021

The revised manuscript was extended at L52:

Due to the combination of multiple observational data sets as input the resulting reanalysis data can outperform individual observational data sets, as shown for e.g. by e.g. Gebremichael et al. (2017) and Zhang et al. (2021) for satellite data.

and L 439:

This study has offered useful insights into the hydrological response of the catchments studied in relation to the selected PRPs. The next step in this field of research would be to evaluate the use

of satellite and other means of remote sensing data to obtain a thorough overview of the available options of handling missing weather station data in Slovenia.

3. The authors site a range of literature on evaluation of reanalysis in Lines 52- 93. However, I don't see a summary of what the reviewed studies did not do, that is compensated by the current study and why reanalysis for Slovenian catchments.  The authors should make this clear in the introduction.

3. We thank reviewer #2 for pointing out a missing structure in the literature review. The intended structure was:

L52 – 76 Focus on literature that validates the rainfall products against observations.

L77  - 93 Review of the literature of reanalysis in hydrological applications. On each reference, besides the location, the temporal resolution is also referred to.

L94 onwards pinpoint gracefully the motivation to validate reanalysis at higher temporal resolutions (daily, hourly), which are not common yet in literature. Also, due to the high spatial variability and intermittency of rainfall a regional validation of reanalysis data is always required/recommended. So, the scientific aim of the study is i) the validation for the temporal resolution of 1 hour and ii) Slovenia as region of interest since it is characterized by large variability in rainfall patterns (e.g., some of the highest rainfall erosivity values in Europe).

We improved the structure of the introduction and added multiple references to point at the research gap this study tries to answer.

4. Why is the ARSO-d data considered as the benchmark? There are other observational precipitation datasets available such as IMERG, MSWEP and GSMaP. The authors should provide more information to claim that ARSO-d data is more accurate than reanalysis products. More details of the on this data should be provided, such as data quality, and the data source, validation etc.

4. The ARSO-d data set is the official data set from the Slovenian Environment Agency (ARSO), which indicates the consideration of the regional knowledge for the spatial interpolation of all rain gauge data from ARSO. Moreover, this dataset does take into consideration all the station-based data available operated by ARSO. IMERG, MSWEP and GSMaP are satellite products (partially in combination with ground-based observations), recommended for regions without ground-based observations (or with low network density), which is not the case for Slovenia on the daily scale. In general, ground observations are considered as 'truth'.

We agree that different rainfall products have different strengths and weaknesses, but a validation of the officially used rainfall data set from the ARSO is beyond the scope of this study. Also, if comparisons with the satellite products recommended by Reviewer #2 indicate deviations, we still would not know which data set to trust more (especially if deviations differ between the satellite products). We stick to ground-based observations as truth, but would implement this discussion in the revised manuscript in order to more clearly address issues raised by the Reviewer #2.

5. How is calibration applied to different forcing?

5. The calibration is carried out following two approaches: 1. Parameters are calibrated for configuration1, then the calibrated parameter set is used for all other configurations; 2. The calibration is re-run for each configuration (L262 -264). Several modifications were made in the manuscript in order to more clearly indicate this.

6. Include the location of the catchments in Table 1 (Lat and Lon), catchment characteristics relevant in the formation and dominant hydrological processes

6. We thank Reviewer #2 for this suggestion. We added the coordinates of the catchments in Table 1. The explanation at L 119 was extended to:

Catchments with larger areas typically receive more precipitation and have higher water storage capacities than smaller catchments, due to larger surface covered. This leads to a lower runoff coefficient and slower runoff response, as water moves through the catchment over longer distances and time periods. Median altitude is another important factor that influences the generation of runoff. Catchments with higher median altitudes generally experience higher precipitation amounts as a result of orographic uplift, which forces moist air to rise and cool, leading to enhanced condensation and precipitation. The percentage of forest area is also a significant factor that affects the generation of runoff. Forested catchments generally have lower runoff coefficients and slower runoff response due to the high rainfall interception rates, and high-water storage capacities of forest soils. The presence of trees also reduces the erosive power of runoff, which leads to lower sediment yields and improved water quality. Mean catchment slope is another factor that contributes in the generation of runoff. Catchments with steeper slopes have higher runoff coefficients and more rapid runoff response due to the reduced infiltration capacity of the soils, and the rapid movement of water down the slope. In contrast, catchments with flatter slopes have lower runoff coefficients and slower runoff response due to the higher infiltration capacity of the soils and the slower movement of water across the landscape.

7. In line 250 – 254, the authors state "The simulation period is split-sampled into….."but do not justify the use of Split Sample method in model calibration. Refer to Arsenault et. al., 2018. The hazards of split-sample validation in hydrological model calibration.

7. We thank the Reviewer #2 for the critical note on split-sampling. Indeed, split-sampling is only recommended if both, calibration and validation period represent similar climate and hence catchment conditions. However, we did not find any issues with the selected periods in terms of very wet or dry periods, so we decided for the 'classical' split-sampling approach. Also, there are references (e.g., Perrin et al., 2003) that support the choice. E.g., the developers of GR4H used the split-sample approach to evaluate the model's performance during the validation period and make adjustments to the model parameters. The authors argue that the split-sample approach is a useful tool for evaluating the performance of hydrological models and avoiding overfitting to the calibration data. They suggest that at least 20% of the data should be reserved for validation to ensure that the model is not overfit to the calibration data and provides a more accurate assessment of the model's performance.

We added the following explanation at L282:

It is very well established that split-sampling is recommended if both calibration and validation periods represent similar climate, and hence catchment conditions. Nonetheless, in the data used for this study there are minimal fluctuations within the selected periods in terms of very wet or dry periods. In addition, different approaches are available as calibration and model evaluation strategies, and some studies support the split-sampling approach (e.g., Perrin et al., 2003). Therefore, a 'classical' split-sampling approach was implemented. However, other methodologies could be tested in future studies.

Perrin, C., Michel, C., & Andreassian, V. (2003). Improvement of a parsimonious model for streamflow simulation. Journal of Hydrology, 279(1-4), 275-289.7.

8.  Line 268 – 273 Please check the font and line spacing

8. We thank Reviewer #2 for pointing to this issue, the format was adapted.

---

## Author Response (AR2)

**Editor**

Dear authors,

While the major concerns raised in the first review round have been cleared, there is a number of minor points to be addressed. I invite you to respond to these points and submit a revised manuscript version.

We thank the editor and both reviewers for their useful suggestions, please see below a detailed response (in green) and the modified parts of the manuscript (in blue) to all the provided comments.

Two editorial comments:
- I recommend using "elevation" instead of "altitude" throughout in text and figures, see McVicar TR & Körner C, 2013. On the use of elevation, altitude, and height in the ecological and climatological literature. Oecologia, 171(2), 335–337. http://dx.doi.org/10.1007/s00442-012-2416-7

We thank the editor for this useful advice, the related parts were rephrased.

- The figures (and text in them) are still slightly blurred, make sure to upload high-resolution images for the final article.

We still think that it is related to pasting it to the word-file, but we will double-check it during the typesetting process.

Please also address the notifications to the authors from review file validation.

Kind regards,
Daniel Viviroli

**Editorial team:**

1. With the next revision, please add the section "Correspondence to:" into the title page of the *.pdf manuscript. Please see more: https://www.hydrology-and-earth-system-sciences.net/submission.html#templates / Technical instructions for MS Word and compatible formats /.

We have added the information as demanded.

2. Please ensure that the colour schemes used in your maps and charts allow readers with colour vision deficiencies to correctly interpret your findings. Please check your figures using the Coblis – Color Blindness Simulator (https://www.color-blindness.com/coblis-color-blindness-simulator/) and revise the colour schemes accordingly.

We are thankful for this advice. The color ramps of two figures have been adjusted to increase readability.

3. Please make bold the captions of all figures for more clarity with the next revision.

Thanks for the advice, all captions should be bold now.

**Reviewer #3**

I have read the revised manuscript as well as the authors' responses to the two referee reports. Overall, the manuscript is suitable for publication in Hydrology and Earth System Sciences, and the major concerns mentioned by the two other referees seem to be cleared out.

We'd like to thank reviewer #3 for this positive evaluation of our study. Detailed response to specific comments is provided below.

However, the following smaller issues and questions should be addressed before publication:

- In your response to point 4 by referee #2 ("Why is the ARSO-d data considered as the benchmark?"), you mention that you also wanted to implement a discussion in the revised manuscript to more clearly address questions in this direction. I may have missed it, but can you indicate where that has been added, or complete if it hasn't been done yet?

We'd like to thank reviewer #3 for pinpointing the matter. The argument for selecting this product for precipitation comparison as the benchmark is provided in the following excerpts:

L170-173: For the validation of the precipitation reanalysis products (PRP), a regionalized daily precipitation data set from the Slovenian Environment Agency (ARSO) is used, from now on referred to as ARSO-d. ARSO-d has a spatial resolution of 1 km raster width length and is available from 01.01.1981 to 31.12.2010. It is based on the regionalization and upscaling of station-based precipitation measurement into spatially and temporally consistent dataset.

L211-212: Hence, for the validation of the PRP the ARSO-d data is used, because it is the only available data set with a sufficient spatial coverage to enable more robust comparisons with the PRP.

L215-217: The PRP are aggregated to daily values to enable the comparison with ARSO-d, since no hourly spatial rainfall product for Slovenia is provided by ARSO.

- Conclusions: Can you make a statement on the extent to which your findings can be applied to other regions, i.e., to what extent can they be generalized, and are limits to that?

We thank the reviewer #3 for this comment. Generalisation is always tricky for rainfall due to its local, regional and hydro-climatological characteristics. We added the following comment in the Conclusions section:

A generalisation of these conclusions is limited due to the regional differences of rainfall processes and the ability of the reanalysis models to represent them. However, the more similar areas of interest are in terms of hydro-climatology, the more likely similar findings can be expected.

- L53: Amend incomplete sentence "… as shown for e.g. by e.g. Gebremichael …"

Thanks, we have completed the sentence now as suggested.

- L61: Do you refer to a commercial context for a specific reason? Otherwise I would rephrase more generally towards applications.

Thanks, we have removed the 'commercial'-part.

- L229: Clarify or replace the term "catchment-intern analyses", it is used only here.

Thanks, we have rephrased it to 'catchment-specific', which is more appropriate here.

- L228: Clarify the "11 years (Jan-Aug) and 10 years (Sep-Dec)" in parentheses: Why these months?

The period of data availability is 01.01.2009-31.08.2019. So there are 11 years available for months Jan-Aug, but only 10 for Sep-Dec. We agree that this information can be misleading and we removed it. The period of data availability remains in the text and we think it will be sufficient for the reader. Thanks.

- L273: You write that "both calibration and validation periods represent similar climate, and hence catchment conditions. " I would not put it that way, as there are other catchment conditions beyond climate.

We thank reviewer #3 for the recommendation and modified this specific part of the manuscript as

It is very well established that split-sampling is recommended if both calibration and validation periods represent similar climate, soil properties and land cover conditions, i.e. consistent catchment conditions over time.

- L275: Please be more specific in this sentence, it sounds rather generic: "In addition, different approaches are available as calibration and model evaluation strategies, and some studies support the split-sampling approach (e.g., Perrin et al., 2003)."

We thank the reviewer #3 for the recommendation and modified this part of the manuscript as:

Nonetheless, in the data used for this study there are minimal fluctuations within the selected periods in terms of very wet or dry periods. In addition, amongst the various methodologies for calibration/validation period selection found in literature, some studies support the split-sampling approach (e.g., Perrin et al., 2003).

-L296: I would mention here in addition that a KGE value of more than -0.41 (i.e., 1-sqrt(2)) indicates model efficiency better than the mean flow benchmark according to Knoben et al. (2019).

We thank reviewer #3 for this literature hint, we added the information as follows:

Knoben et al. (2019) point out that mean flow as benchmark is already outperfrmed with KGE values >-0.41.

- L426: The figure legend seems to be mixed with the main text.

Yes, we have separated it now. Thanks for this remark.

- L440ff: Clarify what you mean by "ad-hoc calibration".

We thank reviewer #3 for the recommendation and added the following segment on L478

Ad-hoc calibration refers to the implementation of the Mitchel algorithm with each new insertion of reanalysis within the model; that is, instead of keeping the model calibrated using observations, its ingrained parameters are re-optimized in order for the reanalysis to converge closer to discharge observations.

- L454: "No significant trends are observable in relation to latitude." Do you use "significant" in a statistical sense? If yes indicate the test done and the p-value used, otherwise change the wording.

We thank reviewer #3 and replaced significant with substantial.

- L476: Complete "distributed" at the end of sentence "PRP is of little added, value compared to a spatially distr."

We have corrected the flaw, thanks for pointing it out.

- L500: Can you give some indications why there are overestimations > 1000 m a.s.l., and why deviations are smaller for catchments below this elevation?

Thanks for this useful comment. We think that the reanalysis models simulate on a certain elevation and got 'corrected'/adapted to the true elevation afterwards. With increasing elevation this correction becomes less accurate, hence higher deviations occur. However, this is an impression from our findings and others from the literature, but further investigations are needed to better evaluate this behavior.

- L503: "the *following* conclusions"?

Thanks, rephrased as suggested.

- L510: "show*s*"

Thanks, the 's' was added.

- L521: You conclude that bias-correction should be implemented, but on L112ff you mention that an advantage of the present research is the examination of raw data. Can you clarify?

We thank reviewer #3 for pointing that concern: in L531 it is emphasized that an evaluation of the reanalysis products that undergo a bias-correction procedure would be beneficial to further assess their applicability in r-r applications.

- Equation 1: Should it be "RC[PRP, n]" and "RC[Obs, n]" instead of "RC[PRP]" and "RC[Obs]" (both in numerator and denominator)?

Yes, indeed we refer to station-based rainfall characteristics. We added the 'i' for all terms.

- Table 1: Use "discharge regime" instead of "water regime" for consistency?

We have rephrased the header as suggested.

- Figure 2: 1) The rivers do not seem to be in alphabetical order, but rather the maxima seem to increase from left to right. Please clarify and amend. 2) I recommend mentioning in the legend that the values are displayed on a logarithmically scaled y-axis. 3) Using different symbols for min/mean/max would improve readability of the figure.

We thank reviewer #3 for his thoughts on the figure, all three suggestions were incorporated.

- Figure 3: Is there a reason for the larger paragraph spacing between Mislinja, Dravinja and Radovna?

No, there was no reason, spacings are uniform now. Thanks for pointing to this issue.

- Figure 6: The figure legend isn't clear to me: "Deviations of areal rainfall extreme values (5- and 50-

year return periods) between ERA5-Land and COSMO-REA6 over all 16 catchments." ? But isn't ARSO-d the reference?

Yes, the disaggregation of ARSO-d using the closest rain gauge is the reference. The caption was misleading, we have changed it to (also for Fig5):

Deviations of areal rainfall extreme values (5- and 50-year return periods) of ERA5-Land and COSMO-REA6 in comparison to observations over all 16 catchments.

- Figure 9: It doesn't seem correct to show negative values in white. KGE has a value range from inf to 1, so values below 0 do not fall outside of that range. Values <0 should rather have an own class in the color bar, and use a color that shows it's a continuum, e.g., dark red.

We thank reviewer #3 for this suggestion, we did as suggested (also for Fig. 12).

- Figure 11: I recommend adding a 1:1 line and making the figure square for better interpretation.

We thank reviewer #3 for this suggestion, the straight line was added.

- Figure 12: This is for the recalibrated model, isn't it? Mention this in legend for clarity.

We have added this information in the caption. Thanks.

Reference:
W. J. M. Knoben, J. E. Freer, and R. A. Woods, 2019: Technical note: Inherent benchmark or not? Comparing Nash-Sutcliffe and Kling-Gupta efficiency scores. Hydrol. Earth Syst. Sci. 23, 4323–4331. doi:10.5194/hess-23-4323-2019.